# A Few Expert Queries Suffices for Sample-Efficient RL with Resets and Linear Value Approximation

**Philip Amortila**$^{\alpha *}$ **Nan Jiang**$^{\alpha}$ **Dhruv Madeka**$^{\beta}$ **Dean P. Foster**$^{\beta}$
$^{\alpha}$University of Illinois, Urbana-Champaign $^{\beta}$Amazon, NYC

## Abstract

The current paper studies sample-efficient Reinforcement Learning (RL) in settings where only the optimal value function is assumed to be linearly-realizable. It has recently been understood that, even under this seemingly strong assumption and access to a generative model, worst-case sample complexities can be prohibitively (i.e., exponentially) large. We investigate the setting where the learner additionally has access to interactive demonstrations from an expert policy, and we present a statistically and computationally efficient algorithm (DELPHI) for blending exploration with expert queries. In particular, DELPHI requires $\tilde{\mathcal{O}}(d)$ expert queries and a $\texttt{poly}(d, H, |\mathcal{A}|, 1/\varepsilon)$ amount of exploratory samples to provably recover an $\varepsilon$-suboptimal policy. Compared to pure RL approaches, this corresponds to an exponential improvement in sample complexity with surprisingly-little expert input. Compared to prior imitation learning (IL) approaches, our required number of expert demonstrations is independent of $H$ and logarithmic in $1/\varepsilon$, whereas all prior work required at least linear factors of both in addition to the same dependence on $d$. Towards establishing the minimal amount of expert queries needed, we show that, in the same setting, any learner whose exploration budget is *polynomially-bounded* (in terms of $d, H$, and $|\mathcal{A}|$) will require *at least* $\tilde{\Omega}(\sqrt{d})$ oracle calls to recover a policy competing with the expert's value function. Under the weaker assumption that the expert's policy is linear (rather than their value function), we show that the lower bound increases to $\Omega(d)$.

## 1 Introduction

Many potential applications of reinforcement learning (RL) have intractably-large state spaces. Thus, we seek provably-correct methods which have statistical and computational requirements that are independent of the size of the state space. This requires some modelling assumptions. One dominating approach has been to introduce *function approximation*, and to posit that the MDP or its value functions are well-represented by the function approximation scheme which is employed. A basic starting point which still lacks comprehensive understanding is the case of linear value function approximation, which models value functions as lying in the span of a known $d$-dimensional feature mapping and asks for methods which have sample complexities that are polynomial only in $d, H$, and possibly $|\mathcal{A}|$ ($H$ and $\mathcal{A}$ are the horizon and action sets of the MDP, respectively). This desideratum was recently understood to be impossible for the "minimal" case where only the optimal value function (or optimal action-value function) is assumed to be linear – i.e. there exist MDPs satisfying this assumption where the statistical complexity of any algorithm will be exponentially large, either in $d$ or in $H$ [WAS21; WSG21; WWK21; FKQR21]. Furthermore, this also holds in the case where the learner is equipped with a generative model (or simulator), allowing them to sample transitions from any state-action pair of the MDP. In recent years much has been said about linear value approximation under stronger assumptions, for example under determinism

---

$^{*}$philipa4@illinois.edu

36th Conference on Neural Information Processing Systems (NeurIPS 2022).

[WR13], linear/low-rank MDPs [JYWJ20; AJSWY20], Bellman-closedness [LSW20; ZLKB20], or the existence of a "core set" [SS20; ZLKB19]. These stronger assumptions can recover polynomial statistical complexities (thought not always computational ones), but are restrictive and oftentimes unrealistic.

In this work, we consider an alternative possibility for recovering polynomial complexities which does not further restrict the class of MDPs under consideration. We do this by assisting the learner with some additional side information about the problem. Specifically, we assume that there is a deterministic expert policy (which need not be the optimal policy) that the learner can query at any state, whereupon they will be informed of the expert's action at that state. Indeed, such information can often be made readily available if we have some form of prior knowledge (or human input) about the problem. Leveraging such expert demonstrations has been studied in interactive imitation learning (IL), with common applications in simulated domains [RGB11; Ros13; RB14; SVGBB17]. As we will see, however, the amount of expert queries required by a pure IL approach is significantly higher than what we need. Since interaction with a (human) expert might be costly, we wish to minimize the burden of the expert by having the learner explore mostly on their own, and only query the expert in a judicious manner. The question asked by this work, then, is:

> *Under linearity of the optimal value function, what is the minimal amount of expert data required for sample/computationally-efficient learning?*

Our main result is the DELPHI algorithm for exploring with an interactive expert. DELPHI assumes that the expert's value function is linear, and that the agent has access to a RESET function which lets them return to the state *most recently seen*. Under these conditions, our method uses surprisingly-few expert queries combined with some modest (polynomial) amount of exploration to recover the expert policy. Formally, DELPHI recovers a policy that is $\varepsilon-$optimal (with respect to the expert policy) with $\mathcal{O}(d\log(B/\varepsilon))$ oracle calls and $\tilde{\mathcal{O}}(\frac{d^2 H^5 |\mathcal{A}| B^4}{\varepsilon^4})$ exploratory samples, where $B$ is a bound on the $\ell_2$-norm of the unknown linear parameter.[2] Thus, our results show that merely $\tilde{\mathcal{O}}(d)$ expert queries enable an **exponential improvement** in sample complexity when compared to RL without expert advice. Furthermore, the number of oracle calls is completely independent of the horizon of the problem and is logarithmic in $1/\varepsilon$, whereas prior work in IL leveraging expert advice requires (at best) linear factors of both in addition to scaling with $d$. We also show that DELPHI is computationally efficient, that it is robust to optimization and misspecification errors, and that it can be extended to the case where the *action*-value function of the expert is linear when the MDP dynamics are deterministic.

Towards establishing the optimality of our algorithm, we study the capabilities of expert-augmented learners which have fixed exploration budgets. More specifically, we ask: what is the minimal number of expert queries *required* by any algorithm which is constrained to a *polynomially-bounded* exploration budget? We show that any polynomially-bounded learner (in terms of $d$, $H$, and $|\mathcal{A}|$) will require at least $\tilde{\Omega}(\sqrt{d})$ oracle calls to recover a policy competing with the expert's value function. In the weaker setting where only the expert's *policy* is linear, we show that this lower bound increases to $\Omega(d)$, matching our upper bound up to logarithmic factors.

The rest of the paper is structured as follows: in Section 2 we review background and present the problem setting. Section 3 describes our algorithm, its guarantee, a sketch of the proof, and discusses some extensions. Section 4 gives the lower bound on the number of expert queries needed. We conclude with an overview of related work and some discussion in Sections 5 and 6.

## 2 Background & Problem Setting

**Notation** We write $\mathrm{Dists}(\mathcal{X})$ for the set of probability measures on some set $\mathcal{X}$. We write $[N] :=$ $\{1, \ldots, N\}$. The *direct product* $\oplus$ corresponds to "concatenating" two vectors, i.e. for any two vectors $u \in \mathbb{R}^n$ and $v \in \mathbb{R}^m$, we have $u \oplus v = (u_1, \ldots, u_n, v_1, \ldots, v_m)^\top \in \mathbb{R}^{n+m}$. We write $\otimes$ for the tensor (or outer) product of two vectors, defined by $u \otimes v = uv^\top \in \mathbb{R}^{n \times m}$, and $\flat(u \otimes v) \in \mathbb{R}^{n \cdot m}$ for the flattening (or vectorization) of said tensor product.

---

[2]The $\tilde{\mathcal{O}}$ notation ignores logarithmic factors.

**MDPs, Policies, and Value Functions**    The typical environment in RL is modelled as an MDP [Put14; Sze10; SB18]. We consider here finite-horizon MDPs, which are specified by a tuple $\mathcal{M} = (\mathcal{S}, \mathcal{A}, \mathcal{R}, \mathcal{P}, H, \mu_0)$, where $\mathcal{S}$ is a state space, $\mathcal{A} = [A]$ is a finite action set, $\mathcal{R} : \mathcal{S} \times \mathcal{A} \to \text{Dists}([0,1])$ is a (bounded) reward distribution function with expectation $r(s, a)$, $\mathcal{P} : \mathcal{S} \times \mathcal{A} \to \text{Dists}(\mathcal{S})$ is the transition distribution function with probability vectors $P(s, a) = [\mathcal{P}(s'|s, a)]_{s' \in \mathcal{S}} \in \mathbb{R}^{|\mathcal{S}|}$, $H \in \mathbb{N}$ is the horizon, and $\mu_0 \in \text{Dists}(\mathcal{S})$ is the starting distribution. Note that we have assumed that the action space is finite, although the state space may be infinite. Without loss of generality we assume that $\mathcal{S}$ is a disjoint union of per-horizon state spaces, i.e. $\mathcal{S} = \cup_{h \in [H]} \mathcal{S}_h$.

A (non-stationary) policy $\pi = (\pi_1, \cdots, \pi_H)$ prescribes a sequence of actions $\pi_h : \mathcal{S}_h \to \text{Dists}(\mathcal{A})$, and its *value function* is $v^\pi(s) = \mathbb{E}[\sum_{h'=h}^{H} r(s_{h'}, a_{h'}) \mid s_h = s, a_{h'} \sim \pi_{h'}(s_{h'})]$, where $s \in \mathcal{S}_h$. The *action-value* function $q^\pi(s, a)$ is defined similarly, save that the first action taken is $a$ and the proceeding actions follow $\pi$. Value functions satisfy the recursive relationship:

$$v^\pi(s) = r(s, \pi) + \langle P(s, \pi), v^\pi(\cdot) \rangle := \mathcal{T}^\pi v^\pi(s) \tag{1}$$

where we have the shorthands $r(s, \pi) = \mathbb{E}_{a \sim \pi(s)}[r(s, a)]$, $P(s, \pi) = \mathbb{E}_{a \sim \pi(s)}[P(s, a)]$, and the Bellman operator $\mathcal{T}^\pi(\cdot) := r(s, \pi) + \langle P(s, \pi), (\cdot) \rangle$. The Bellman operator has $v^\pi$ as its unique fixed point. The optimal policy is written as $\pi^\star$, and its value function is denoted as $v^\star := v^{\pi^\star}$. The objective is to find a $\pi$ maximizing $v^\pi(\mu_0) := \mathbb{E}_{s_0 \sim \mu_0}[v^\pi(s_0)]$.

**Function Approximation With an Interactive Expert**    In the RL setting, the MDP is unknown and must be explored. As stated in the introduction, we seek sample complexities which are independent of the number of states. This is evidently not possible without further assumptions. To assist the learner, we assume that the agent has further access to an *oracle*, which, upon being queried, returns the action of an *expert* policy $\pi^\circ$ for the state. The expert policy need not be the optimal policy. We will assume that $\pi^\circ$ is deterministic, which is satisfied for example when $\pi^\circ = \pi^\star$ or when $\pi^\circ$ is the greedy policy of some value function.

**Assumption 2.1** (Interactive expert)**.** *There is an oracle which can be queried at the current state $s$, which returns the action $\pi^\circ(s)$. Syntactically, the oracle is queried via the* ORACLE$(s)$ *function.*

We note that the oracle can only be queried at the state the learner is currently visiting. The objective of the learner is to recover a policy which competes with the expert policy, namely a $\hat{\pi}$ such that:

$$v^{\hat{\pi}}(\mu_0) \geq v^\circ(\mu_0) - \varepsilon \quad \text{with probability } \geq 1 - \delta, \tag{2}$$

where $v^\circ := v^{\pi^\circ}$ is the value function of the expert. In the sequel we refer to a policy satisfying Equation 2 as $\varepsilon$-*optimal*.

Our next assumption concerns the interplay of the expert policy with linear value approximation. Namely, we assume that the expert policy has a value function which is linear in a set of known features.

**Assumption 2.2** ($v^\circ$-linearity, with bounded features)**.** *The value function $v^\circ$ of the expert is linear with known features $\varphi \in \mathbb{R}^d$, i.e.*

$$v^\circ(s) = \langle \varphi(s), \theta^\circ \rangle, \ \forall s \in \mathcal{S}, \tag{3}$$

*for some unknown $\theta^\circ \in \mathbb{R}^d$. We further assume that $\|\varphi(s)\|_2 \leq 1 \ \forall s$ and that $\|\theta^\circ\|_2 \leq B$ for some known $B \in \mathbb{R}^d$.*

This assumption can be extended to the case of (slight) misspecification, as will be shown in Subsection 3.2. Our last assumption is that the agent has the ability to "reset" to the state most recently seen.

**Assumption 2.3** (Resets)**.** *After experiencing a transition $(s, a, r, s')$ in the MDP, the agent can return to the state $s$. Syntactically, this is done via the* RESET$()$ *function.*

Our RESET assumption is weaker than the full generative model access [KMN02] or the "local" simulator setting [WAJAYJS21; LCCGW21; HLYAYS22] which has appeared in prior works. As noted in the introduction, existing exponential lower bounds entail that Assumptions 2.2 and 2.3 together are not enough to enable sample-efficient learning. Thus, any efficient learner which is provided with the above assumptions (2.1, 2.2, and 2.3) must necessarily make use of the expert.

# 3 The DELPHI algorithm

We are ready to describe our approach and give the main result. We begin by supposing that the starting distribution is deterministic (we will see later that this comes at no loss of generality).

**Theorem 3.1.** *Suppose Assumptions 2.1, 2.3, and 2.2 hold. Then the DELPHI algorithm will recover a policy $\hat{\pi}$ such that $v^{\hat{\pi}}(s_0) \geq v^{\circ}(s_0) - \varepsilon$ with probability $\geq 1 - \delta$, using $\mathcal{O}(d \ln(B/\varepsilon))$ oracle calls and $\tilde{\mathcal{O}}(\frac{d^2 H^5 A B^4}{\varepsilon^4})$ interactions with the MDP. Furthermore this algorithm is computationally efficient.*

The pseudo-code for DELPHI is given in Algorithm 1, which uses Algorithm 2 (`measureTD`) as a sub-routine to measure expectations. DELPHI will solve for a parameter $\hat{\theta}$ such that an induced policy $\pi_{\hat{\theta}}$ (defined via Eq. (4)) will compete with the expert's value function. We note that deploying the policy $\pi_{\hat{\theta}}$ also requires the RESET functionality, since expectations must be estimated from a small number of samples at each state encountered. This is a consequence of our assumption that $v^{\circ}$ is linear (rather than, e.g., $q^{\circ}$ or the MDP itself), since selecting actions based on state-value functions will always require one-step look-aheads.

**Intuition for DELPHI** Recall that the expert policy $\pi^{\circ}$ satisfies $v^{\circ} = \mathcal{T}^{\pi^{\circ}} v^{\circ}$, and that this fixed point is unique. We say that a candidate value function is *consistent* at a state $s$ if $v(s) = \mathcal{T}^{\pi^{\circ}} v(s)$. Note that we need consistency to hold globally (i.e. at all states) in order to ensure that $v = v^{\pi^{\circ}}$. Our methodology is based on ensuring that consistency on a small number of well-chosen states will guarantee global consistency.

DELPHI is inspired by a recent algorithm of [WAJAYJS21] called TensorPlan. As in TensorPlan, DELPHI proceeds via a "guess and check" procedure: at every iteration, we pick the *optimistic* linear parameter which is consistent on the past expert data that we have seen. We then check whether this choice of parameter is globally consistent, by playing $n_{\texttt{rollout}}$ rollouts of length $H$ with a policy derived from the parameter. More specifically, for any $\theta$ the policy $\pi_{\theta}$ takes the form

$$\pi_{\theta}(s) = \operatorname{argmin}_a \left| \left( \hat{r}(s,a) + \hat{\mathbb{E}}_{s,a}[v_{\theta}(s')] \right) - v_{\theta}(s) \right|, \tag{4}$$

where $v_{\theta}(\cdot) := \langle \varphi(\cdot), \theta \rangle$, and $\hat{r}(s,a)$ and $\hat{\mathbb{E}}_{s' \sim P(s,a)}$ are estimated via the RESET function.

After a certain number of rollouts, one of two things happen: either this policy encounters a state where there is no consistent action (i.e. the above minimum has a strictly positive value), or we only encounter states that are consistent.[3] In the first case, we query the oracle for its expert action and use the transition for that action to update the parameter set. In the second case, we derive (cf. Lemma 3.7) that if no inconsistencies are observed for several rollouts, then our "virtual value" $v_{\theta}$ is close to the true value under $\pi_{\theta}$ (i.e., $v^{\pi_{\theta}}$). Using that $\theta$ was optimistic (i.e. $v_{\theta}(s_0) \geq v_{\theta^{\circ}}(s_0)$), this implies that we are optimal.

The only thing left to argue is that the number of iterations (i.e. the number of times that we can continue finding new parameters which are not globally consistent) is small. Using linearity of $v^{\circ}$, it turns out roughly $d$ inconsistencies are sufficient for this. To see this, note that we can re-write the Bellman equation for any $v_{\theta}(\cdot) = \langle \varphi(\cdot), \theta \rangle$ as:

$$v_{\theta}(s) = \mathcal{T}^{\pi^{\circ}} v_{\theta}(s) \iff 0 = r(s, \pi^{\circ}(s)) + \langle \mathbb{E}_{s' \sim P(s, \pi^{\circ})}[\varphi(s')] - \varphi(s), \theta \rangle$$
$$\iff 0 = \langle \Delta_{s, \pi^{\circ}(s)}, 1 \oplus \theta \rangle, \tag{5}$$

where we have introduced the notation $\Delta_{s,a} := r(s,a) \oplus (\mathbb{E}[\varphi(s')] - \varphi(s))$ and used linearity of expectation, linearity of inner products, the definition of $v_{\theta}$, and the definition of the direct product. We call the vector $\Delta_{s,a}$ the *temporal difference* (TD) vector for $(s,a)$. Equation (5) is precisely an orthogonality constraint in $d + 1$ dimensions. Since the parameter $\theta_t$ which is chosen at time $t$ was consistent on past data, it is orthogonal to the previous $t - 1$ TD vectors which have been generated from interactions with the oracle. If we happen to find a state which has no consistent action, then the TD vector corresponding to the expert action at that state must not be in the span of the previous expert TD vectors (otherwise it would be consistent). It follows that the iteration complexity is at most $d + 1$, since there are at most $d + 1$ linearly independent vectors in $\mathbb{R}^{d+1}$. We use the Eluder dimension [RVR13] to generalize this argument to the case where the expectations are estimated.

The next section solidifies the above intuition and sketches the proof more formally.

---

[3]In reality, due to sampling errors, we tolerate some error during the consistency checks.

---

**Algorithm 1** DELPHI

---

1: **Inputs:** $s_0, \varphi,$ `sub-optimality` $\varepsilon_{\texttt{target}},$ `confidence` $\delta,$ `parameter bound` $B$
2: $\Theta_0 \leftarrow \texttt{Ball}_{\ell_2}(B)$           $\triangleright \Theta_t$ : current consistent parameters
3: Initialize $E_d,\ n_{\texttt{eval}}, n_{\texttt{rollout}},$ and $\varepsilon_{\texttt{tol}}$ via Equations (9), (12), (10), (15)
4: **for** $t = 1$ to $E_d + 1$ **do**
5:   Pick $\theta_t \in \text{argmax}_{\theta \in \Theta_{t-1}} \left( v_\theta(s_0) := \theta^\top \varphi(s_0) \right)$    $\triangleright$ Optimistic choice over $\Theta_{t-1}$
6:   `consistent` $\leftarrow$ `true`
7:   **for** $m = 1$ to $n_{\texttt{rollout}}$ **do**     $\triangleright\ n_{\texttt{rollout}}$ number of rollouts with $\theta_t$-induced policy
8:    $S_{t,m,h} = s_0$               $\triangleright$ Initialize rollout
9:    **for** $h = 1$ to $H$ **do**
10:     **for** $a \in [A]$ **do**            $\triangleright$ For each action
11:      $\hat{\Delta}_{S_{t,m,h},a} \leftarrow \texttt{measureTD}(S_{t,m,h}, a, n_{\texttt{eval}})$   $\triangleright$ Measure TD vector at $(s,a)$
12:     **end for**
13:     **if** $\min_a \left| \langle \hat{\Delta}_{S_{t,m,h},a}, 1 \oplus \theta_t \rangle \right| > \varepsilon_{\texttt{tol}}$ **then**     $\triangleright$ No consistent action
14:      `consistent` $\leftarrow$ `false`
15:      $a_t^\circ \leftarrow \text{ORACLE}(S_{t,m,h})$      $\triangleright$ Query oracle for $\pi^\circ(S_{t,m,h})$
16:      $\tilde{\Delta}_{S_{t,m,h},a_t^\circ} \leftarrow \texttt{measureTD}(S_{t,m,h}, a_t^\circ, 4E_d n_{\texttt{eval}})$    $\triangleright$ Refined data
17:      $\Theta_t \leftarrow \Theta_{t-1} \cap \{\theta \mid |\langle \tilde{\Delta}_{S_{t,m,h},a_t^\circ}, 1 \oplus \theta \rangle| \le \varepsilon_{\texttt{tol}}\}$   $\triangleright$ New admissible parameters
18:      Exit current iteration, $t \leftarrow t+1$, Goto Line 5.
19:     **end if**
20:     $A_{t,m,h} \leftarrow \text{argmin}_{a \in [A]} \left| \langle \hat{\Delta}_{S_{t,m,h},a}, 1 \oplus \theta_t \rangle \right|$    $\triangleright$ Else consistent, keep playing
21:     Play $A_{t,m,h}$, get $R_{t,m,h}, S_{t,m,h+1} \sim$ MDP     $\triangleright$ Roll forward
22:    **end for**
23:   **end for**
24:   **if** `consistent == true` **then**
25:    **return** $\theta_t$       $\triangleright$ No inconsistency for $m$ rollouts $\implies$ success
26:   **end if**
27: **end for**
28: **return** $\theta_{E_d+1}$

---

---

**Algorithm 2** `measureTD`

---

1: **Inputs:** $s, a, \varphi(\cdot), n, \text{RESET}()$
2: **for** $i = 1$ to $n$ **do**
3:   Play action $a$ at $s$, receive sample $R_l$ and $S_l'$ from MDP
4:   $\Delta_i \leftarrow R_l \oplus (\varphi(S_l') - \varphi(s))$
5:   $\text{RESET}()$
6: **end for**
7: **return** $\hat{\Delta}_{s,a} := \frac{1}{n} \sum_{i \in [n]} \Delta_i$

---

### 3.1 Proof sketch

The full proof comes in 4 parts. The proofs for all Lemmas and the precise values of all hyperparameters are given in Appendix A.

1. Lemmas 3.2 and 3.3 gives concentration bounds which establish that, with high probability, the measurements $\hat{\Delta}$ (Line 11) and $\tilde{\Delta}$ (Line 16) concentrate to the average $\Delta_{s,a} = \mathbb{E}_{R(s,a)} r \oplus \left( \mathbb{E}_{P(s,a)} \varphi(s') - \varphi(s) \right)$.

2. Lemma 3.4 establishes that, with high probability, the true optimal parameter $\theta^\circ$ is not eliminated from $\Theta_t$ for any parameter set that is encountered. It follows (Lemma 3.5) by optimism that $v_{\theta_t}(s_0) \ge v^\circ(s_0)$ with high probability, where $\theta_t$ is the parameter chosen at time $t$.

3. Lemma 3.6 establishes an iteration bound: the algorithm will terminate after at most $t = E_d$ iterations of the outermost loop (and thus after at most $E_d$ oracle queries). The quantity $E_d$ happens to be the Eluder dimension of our linear function class.

4. Lastly, Lemma 3.7 establishes that if $n_{\texttt{rollout}}$ number of rollouts occur without observing a consistency break, then the virtual value ($v_\theta(s_0)$) must be roughly equal to the true value under the executed policy ($v^{\pi_\theta}(s_0)$). Theorem 3.1 combines all the ingredients to conclude the proof.

**Part 1: Concentration bounds**   Recall that $\Delta_{s,a} := r(s,a) \oplus (\mathbb{E}[\varphi(s')] - \varphi(s))$ is the true TD vector, $\hat{\Delta}$ is the estimated TD vectors obtained with $n_{\texttt{eval}}$ samples (in Line 11), and $\tilde{\Delta}$ is the "refined data" obtained with $4E_d n_{\texttt{eval}}$ samples (in Line 16). The following lemmas establish concentration of $\hat{\Delta}$ and $\tilde{\Delta}$ to the true TD vector.

**Lemma 3.2** (Concentration of $\hat{\Delta}_{s,a}$ (Line 11)). *For any $s, a \in \mathcal{S} \times \mathcal{A}$ that is observed throughout the execution* DELPHI *, with $n_{\texttt{eval}}$ samples in Line 11, we have that with probability $\geq 1 - \delta$, $\left\|\hat{\Delta}_{s,a} - \Delta_{s,a}\right\|_\infty \leq \varepsilon_{\texttt{eval}}$ and thus that $\langle 1 \oplus \theta, \hat{\Delta}_{s,a} - \Delta_{s,a}\rangle \leq \bar{\varepsilon}_{\texttt{eval}}$.*

**Lemma 3.3** ($\tilde{\Delta}_{s,a}$ concentrates even more (Line 16)). *Similarly, for all $s, a$ where we call the oracle, with probability $1 - \delta$, we have $\left\|\tilde{\Delta}_{s,a} - \Delta_{s,a}\right\|_\infty \leq \varepsilon_{\texttt{eval}}/(2\sqrt{E_d})$, and thus, $\forall \theta \in Ball_{\ell_2}(B)$, $|\langle 1 \oplus \theta, \tilde{\Delta}_{s,a} - \Delta_{s,a}\rangle| \leq \bar{\varepsilon}_{\texttt{eval}}/(2\sqrt{E_d})$.*

**Part 2: Optimism**   This part shows that (with high probability) the true optimal parameter is not eliminated from the version space, and thus by optimism that the predicted value $v_t$ upper bounds $v^\circ$.

**Lemma 3.4** ($\theta^\circ$ not eliminated). *With probability $\geq 1 - \delta$, $\theta^\circ \in \Theta_t$ for all iterations $t \in [E_d + 1]$.*

**Lemma 3.5** (Optimism). *Under the event of Lemma 3.4, we have $v_t(s_0) \geq v^\circ(s_0)$, $\forall t \in [E_d]$.*

**Part 3: Iteration bound**   To bound the iteration complexity of our algorithm, we use the notion of *Eluder* dimension [RVR13]. Loosely, the Eluder dimension with respect to some target function is the longest sequence of points $(x_i)$ such that there exists functions differing from the target function on $x_i$ but which correctly fit it on $x_1, \ldots, x_{i-1}$. A formal definition is provided in Appendix A. We will use the result that the Eluder dimension of linear functions is $\mathcal{O}(d\ln(B/\varepsilon))$ [RVR13; LKFS21].

**Lemma 3.6** (Iteration Complexity). *With probability $\geq 1 - 2\delta$, the iteration complexity of the algorithm is at most the Eluder dimension at scale $\bar{\varepsilon}_{\texttt{eval}}$, i.e. $E_d = \mathcal{O}(d\ln(B/\bar{\varepsilon}_{\texttt{eval}}))$.*

**Part 4: Consistency, and putting everything together**   The last thing to show is that if we are consistent at all states for several rollouts then the virtual value $v_\theta$ will be close to the value of the $v^{\pi_\theta}$.

**Lemma 3.7** (Consistency $\implies$ accurate prediction). *If $m$ rollouts have occured without any inconsistencies (i.e., the if statement of Line 13 never gets triggered), then $v^{\pi_\theta}(s_0) > v_\theta(s_0) - 5H\bar{\varepsilon}_{\texttt{eval}} - \varepsilon_{\texttt{roll}}$ with probability $\geq 1 - 3\delta$.*

We are now ready to put everything together.

*Proof (of Theorem 3.1).* Assume all events introduced so far (i.e. the events in Lemma 3.2, Lemma 3.3, and Lemma 3.7). Together these happen with probability $\geq 1 - 3\delta$, so we can re-define $\delta \mapsto \delta/3$ such that the events happen together with probability $\geq 1 - \delta$ (this only increases factors inside logarithms by 3). By Lemma 3.7, we have:

$$
\begin{aligned}
v^{\pi_\theta}(s_0) &\geq v_\theta(s_0) - 5H\bar{\varepsilon}_{\texttt{eval}} - \varepsilon_{\texttt{roll}} \\
&\geq v_\theta(s_0) - \varepsilon_{\texttt{target}} \\
&\geq v^{\pi^\circ}(s_0) - \varepsilon_{\texttt{target}},
\end{aligned}
\tag{6}
$$

where the second step follows from plugging in the definitions of $n_{\texttt{eval}}$ (Eq. (12)), $n_{\texttt{rollout}}$ (Eq. (10)), $\bar{\varepsilon}_{\texttt{eval}}$, (Eq. (13)) and $\varepsilon_{\texttt{roll}}$ (Eq. (16)), and the final step follows by optimism (Lemma 3.5). The total sample complexity of our algorithm is: $E_d = \tilde{\mathcal{O}}(d)$ oracle calls, and $N n_{\texttt{eval}} = (E_d + 1)H n_{\texttt{rollout}} A n_{\texttt{eval}} = \tilde{\mathcal{O}}(\frac{d^2 H^5 A B^4}{\varepsilon_{\texttt{target}}^4})$ exploration cost. The last claim to verify is that of computational efficiency. We note that the only computationally intensive step from iteration is Line 5, i.e. the optimization problem corresponding to the optimistic choice over the parameter set $\max_{\theta \in \Theta_{t-1}} \theta^\top \varphi(s_0)$. This is readily seen to be a convex program, since the objective is a linear

function and the constraint set is a convex set in $\mathbb{R}^d$ (it is initialized at $\Theta_0 = \texttt{Ball}_{\ell_2}(B)$ and every update intersects it with half-spaces (Line 17)). Besides the $\ell_2$-norm constraint, all other constraints can be represented with a linear program, since each absolute value constraint can be split into two inequalities. By Lemma 3.6 the number of constraints will be at most $2E_d = \tilde{\mathcal{O}}(d)$ (and these can easily be converted to standard form). Thus, there are a plethora of polynomial-time methods for solving this convex program [BBV04; Bub+15].[4] $\qquad\square$

## 3.2 Extensions

In this section, we show that DELPHI can be extended to work with optimization errors, under stochastic starting distributions, with misspecification, and with linear $q^\circ$ whenever dynamics are deterministic.

**Optimization errors and stochastic start state**   If the optimistic program can only be solved upto accuracy $\varepsilon_{\texttt{opt}} \leq c\varepsilon_{\texttt{target}}$ (for $c \in [0,1)$) at each iteration, then Lemma 3.5 becomes $v_t(s_0) \geq v^\circ(s_0) - \varepsilon_{\texttt{opt}}$ and this only appears in Eq. (6). We then need $5H\bar{\varepsilon}_{\texttt{eval}} + \varepsilon_{\texttt{roll}} \leq (1-c)\varepsilon_{\texttt{target}}$, which is achieved by increasing $n_{\texttt{rollout}}$ and $n_{\texttt{eval}}$ by a factor of $1/(1-c)^2$. This increases the final exploration cost by $1/(1-c)^4$. For stochastic starting states, we simply work with $v_\theta(\mu_0) = \mathbb{E}_{s_0 \sim \mu_0}[v_\theta(s_0)]$ (resp. $v^\circ(\mu_0)$) wherever $v_\theta(s_0)$ (resp. $v^\circ(s_0)$) previously appeared. The "starting feature" $\mathbb{E}_{s_0 \sim \mu_0}[\varphi(s_0)]$ must be estimated from samples, which is then used for the optimistic program in Line 5 with $\varphi(s_0)$ replaced by this expectation. The error is easily bounded as before by Hoeffding's inequality, and will simply propagate additively through the proof.

**Misspecified value functions and innacurate simulators**   DELPHI inherits some robustness properties from TensorPlan. Namely, with a constant increase in exploration cost, DELPHI continues to work under errors in the modelling assumptions. The first case is where the expert value function is not linear but rather is approximately linear up to some uniform error. Formally, we say that the MDP is $\eta$-*misspecified* for the expert policy $\pi^\circ$ and the feature map $\varphi$ if there exists $\theta^\circ$ such that $\sup_s |v^\circ(s) - \langle \varphi(s), \theta^\circ \rangle| \leq \eta$. The second case is where the simulator itself is flawed. Formally, we say that the simulator is $\lambda$-*innacurate* if a transition $(r, s')$ from any state-action pair $(s, a)$ of the MDP is instead observed as $(\Pi(r + \lambda_{s,a}), s')$, where $\Pi$ is the projection onto $[0,1]$ and $\lambda_{s,a}$ is a constant uniformly bounded by $\lambda$. The following result (proved in Appendix A.5) states that DELPHI can tolerate misspecification or simulator inaccuracies of order roughly $\frac{\bar{\varepsilon}_{\texttt{eval}}}{\sqrt{E_d}} = \mathcal{O}(\frac{1}{H\sqrt{d}})$.

**Theorem 3.8** (DELPHI with misspecification).   *Redefine $n'_{\texttt{eval}} = 4n_{\texttt{eval}}$ (Eq. 12) and all subsequent hyperparameters which depend on $n'_{\texttt{eval}}$. Then we have that, for all MDPs that are at most $\frac{\bar{\varepsilon}_{\texttt{eval}}}{8\sqrt{E_d}}$- misspecified or for all simulators that are at most $\frac{\bar{\varepsilon}_{\texttt{eval}}}{4\sqrt{E_d}}$-innacurate, the conclusions of Theorem 3.1 continue to hold when running DELPHI with the new hyperparameters.*

**$q^\circ$-linearity, in deterministic dynamics**   Rather than working with the Bellman equation $v_\theta(s) = \mathcal{T}^{\pi^\circ} v_\theta(s)$ we work with $q_\theta(s, a) = \mathcal{T}^{\pi^\circ} q_\theta(s, a)$, which can be linearized similarly to Eq. (5). Namely:

$$q_\theta(s, a) = \mathcal{T}^{\pi^\circ} q_\theta(s, a) \iff 0 = \langle r(s,a) \oplus (\mathbb{E}[\varphi(s', \pi^\circ(s))] - \varphi(s,a)), 1 \oplus \theta \rangle \qquad (7)$$

This derivation holds generally, although to be able to speak of consistency at $(s, a)$ with respect to a next action $a'$, we now assume deterministic dynamics, so that the above becomes

$$0 = \langle r(s,a) \oplus (\varphi(s', a') - \varphi(s,a)), 1 \oplus \theta \rangle,$$

where $s'$ is the unique successor of $(s, a)$ and $a'$ is the action that we are checking consistency for. The algorithm proceeds as before, except rather than checking all actions at a given state (Line 13), we check all proceeding actions $a'$ against the current $(s, a)$, and play the one with the smallest TD error (analogously to Line 20 of Algorithm 1).

---

[4]In fact, replacing $\Theta_0$ with an $\ell_\infty$ constraint will only incur a $d^2$ factor to the exploration cost and log factors to the oracle cost (since we can think of $B \mapsto \sqrt{d}B$), but Line 5 will then be a fully linear program.

# 4 How many oracle calls are necessary?

Is DELPHI optimal in terms of its number of oracle calls? To answer this question, we must argue that no algorithm can find an $\varepsilon-$optimal solution with less than $\Omega(d)$ expert queries. As stated, we are competing with agents which can (for example) exhaustively search the state-space and do not refer to the expert at all. Thus, an *exploration budget* must be imposed (formally, a maximal amount of allowed interaction with the MDP, excluding oracle calls). To make matters more interesting, we set this cap to be *any* polynomial amount, resulting in the following question:

> *Under Assumptions 2.1, 2.2, and 2.3, what is the minimal amount of expert queries needed for any algorithm with a $poly(d, H, A, \frac{1}{\varepsilon})$ exploration budget to find an $\varepsilon-$optimal solution?*

We firsly note that, measured in the worst-case, this minimal amount of expert queries is strictly positive since there exists MDPs satisfying $v^\star$-linearity for which no algorithm can return a sound solution with $poly(d, H, A, \frac{1}{\varepsilon})$ queries [WAS21; WSG21]. Secondly, while it was possible to restrict ourselves simply to learners which have the same exploration requirements as DELPHI , we opted to study algorithms with arbitrary polynomial exploration budgets, since it is a more fundamental question about the limits of exploration and the benefits of expert advice.

Note that any solution to this question must, a priori, have an exponential sample complexity for pure RL (otherwise the agent does not need to resort to the expert). Interestingly, most constructions which exhibit exponential lower bounds for linearly-realizable RL can be solved with a single query from the oracle (e.g., [WAS21; WWK21]). These constructions rely on having an exponentially large action set with a single correct action that effectively solves the MDP. Our main lower bound comes from extending the recent lower bound of [WSG21], which is also the only known construction for an exponential lower bound which has a polynomial action set (rather than exponential). Our result is that at least $\tilde{\Omega}(\sqrt{d})$ oracle calls are necessary:

**Theorem 4.1.** *There exists a family of MDPs satisfying 2.1, 2.2, and 2.3, such that any algorithm with $poly(d, H, A)$ exploration budget will need at least $\tilde{\Omega}(\sqrt{d})$ oracle calls to recover a policy such that $v^{\hat\pi}(s_0) \geq v^\circ(s_0) - 0.01$.*

Our second lower bound considers an alternative assumption, which instead posits that the expert *policy* is linear. Formally, this assumption says:

**Assumption 4.2** ($\pi^\circ$-linearity, with bounded features)**.** *The policy $\pi^\circ$ of the expert is linear with known features $\varphi : \mathcal{S} \times \mathcal{A} \to \mathbb{R}^d$, i.e.*

$$\pi^\circ(s) \in \operatorname{argmax}_a \langle \varphi(s, a), \theta^\circ \rangle, \ \forall s \in \mathcal{S}, \tag{8}$$

*for some unknown $\theta^\circ \in \mathbb{R}^d \setminus \{0\}$.[5] We further assume that $\|\varphi(s)\|_2 \leq 1 \, \forall s$ and that $\|\theta^\circ\|_2 \leq B$ for some known $B \in \mathbb{R}^d$.*

When $\pi^\circ = \pi^\star$, it is easy to see that this assumption is implied by the assumption that $q^\star$ is linearly-realizable. In general however, $q^\circ$-linearity does not imply that $\pi^\circ$ is linear (see Appendix C for an example), although it does imply that the greedy policy derived from $q^\circ$ is linear. We give a lower bound for this case which matches the upper bound of DELPHI up to logarithmic factors.

**Theorem 4.3.** *There exists a family of MDPs satisfying assumptions 2.1, 4.2, and 2.3 such that any algorithm with $poly(d, H, A)$ exploration budget will need at least $\Omega(d)$ oracle calls to recover a policy such that $v^{\hat\pi}(s_0) \geq v^\circ(s_0) - 0.01$.*

**Intuition for the lower bound**   Theorems 4.1 and 4.3 use the same MDP construction but with different features. We give some intuition for the MDP construction which is used, but due to its intricacy a full description (and the information-theoretic proof) are deferred to Appendix B. Loosely, the learner has to find a hidden hypercube vector $s^\star \in \{\pm 1\}^p$. The action space is $\mathcal{A} = [p]$, and each action corresponds to flipping one of the bits of the current vector. The MDP has $K$ "phases" which each correspond to $p$ bit flips (thus $H \approx Kp$). A linear reward is given only if a sufficiently small neighborhood of $s^\star$ is reached, and the reward (thus the value) will decay geometrically in each subsequent phase that the neighborhood is not reached. Intuitively, the oracle needs to be used

---

[5] We exclude 0 since otherwise this would simply be the class of all policies.

$\approx p$ times, since each oracle calls only reveals one action, and thus one bit of the optimal vector $s^\star$. The reason that this results in a $\tilde{\Omega}(\sqrt{d})$ lower bound (rather than $\tilde{\Omega}(d)$) is that the value function will experience a *scale transition* when going from states where $s^\star$ is reachable given the remaining steps in the current phase to states where $s^\star$ is no longer reachable. As just described, the value (and thus the features) will be one order of magnitude smaller in this latter portion of the state space. As this would betray the location of the secret parameter, the value function is instead augmented to be quadratic in $p$ (roughly, the product of the distance achieved at the end of this phase and that of the next phase), which requires that $p \approx \sqrt{d}$ in order to observe linearity. On the other hand, the lower bound for $\pi^\circ$-linearity (Theorem 4.3) can remain linear in $d$, since the definition (Eq. 8) is *scale-insensitive*.

Closing the gap between our upper bound of $\tilde{\mathcal{O}}(d)$ and our lower bound of $\tilde{\Omega}(\sqrt{d})$ remains a challenging but interesting question. In finite horizons, with access to a generative model or a RESET method, we suspect that the only mechanism for creating a hard MDP is by geometrically decaying the maximum possible value for each stage, such that at the final stage of the MDP the (random) reward becomes exponentially small in $H$ (that is the approach taken here and in [WAS21; WSG21]).[6] If the geometric decaying happens in "phases" then this implies that $\Omega(1)$ of the value is located in the first "phase". The tension, then, is to have such a construction while hiding this large value, and forcing the learner to rely on several oracle calls to find it. Prior constructions [WAS21; WWK21] have hid the large initial value by choosing an exponentially large action set, although as discussed above these examples are solved with a single query of the expert oracle. Extending the "needle in a haystack" to occur over multiple decisions (cf. the "phases" used above) leads to increased oracle requirements, although due to the scale transition phenomenon observed above it is far from clear how to do this with phases of length $\approx d$. The question thus boils down to: is geometric value reduction *necessary* for exponential lower bounds in this setting, and if so can we avoid the scale transition problem? The situation is likely to be different in the *online RL* setting. For example, the construction of [WWK21] extends [WAS21] to the online setting, and does not need to decay rewards but instead adds an $\Omega(1)$ probability of death at every transition. This mechanism evidently does not work when the agent has resets, and since it is not known whether DELPHI can be extended to the online setting we opted to keep the settings consistent between our upper bound and our lower bound.

## 5   Related works

The closest body of work to our setting is the field of *interactive* IL. As in our setting, interactive IL considers the case where the learner has access to an expert oracle that can be queried adaptively. It differs from our setting, however, since traditionally in IL the learner does not observe reward information. We further differ from the IL setting since we consider value function approximation rather than general policy classes, and since we assume access to a RESET function. Despite that many demonstrations of interactive IL occur in simulated domains [RGB11; Ros13; RB14], the benefits of this feature have not previously been studied. Our assumption of $v^\circ$ linearity entails that many IL methods are not directly applicable. Indeed, the policy $\pi^\circ$ itself does not need to be linear (despite that $v^\circ$ is), so it is unclear which policy class to use for those algorithms. Assuming for the sake of comparison that linear policies can be used, IL methods would still obtain worse oracle rates. Indeed, using results from [AJKS19], Behaviour Cloning (for the passive case) or AggreVaTe [RB14; SVGBB17] (for the interactive case) have worst-case errors $\mathcal{O}\big(\frac{1}{(1-\gamma)^2}\sqrt{\frac{d\ln(1/\delta)}{N}}\big)$ for discounted MDPs, which roughly translates to an oracle complexity of $N = \mathcal{O}(dH^4/\varepsilon^2)$ when using the standard reduction $H \mapsto (1-\gamma)^{-1}$. This is in sharp contrast to our $\tilde{\mathcal{O}}(d)$ oracle calls, which in independent of $H$ and logarithmic in $1/\varepsilon$, and demonstrates the improvement due to exploration with the help of value-function approximation. Beyond these approaches, another intuitive method would be to perform regression by doing a Monte Carlo estimation for the value of $v^\circ(s)$ for each $s$ along a certain "good" set of features. This would require collecting rollouts from those states, which will again introduce a factor of $H$ in the number of oracle queries. Our algorithm instead finds a set of state-action pairs where the Temporal difference (TD) errors (which we represent as vectors) span orthogonal directions. These can be estimated with a single transition, and this "local fitting" approach is novel to the IL literature and avoids the factors of $H$ and $1/\varepsilon$ from previous works.

---

[6]In particular, an exponentially small gap is necessary, since backwards induction-type methods are possible when $q^\star$ is linearly-realizable and have sample complexities scaling with the inverse gap [DKWY19; DLMW20].

In terms of other linear structure in IL, the works of [AN04; SS07] assume a known transition function and unknown linear rewards, and derive expert complexities of $\mathcal{O}\left(\frac{dH^2 \log(dH/\varepsilon)}{\varepsilon^2}\right)$ and $\mathcal{O}(\frac{H^2 \log(d)}{\varepsilon^2})$ respectively, but the algorithms involve (tabular) planning in MDPs and thus are not computationally efficient. Most relevant is the recent work of [RHYLJR21], which, in the reward-free case, assumes that the expert policy is linear (i.e., our Assumption 4.2). A sample complexity of $\tilde{\mathcal{O}}(dH/\varepsilon)$ is shown for Behaviour Cloning in this case, though no lower bound is given.

On the technical side, the DELPHI algorithm is inspired by a recent algorithm called TensorPlan [WAJAYJS21]. TensorPlan works for pure RL under Assumptions 2.2 and 2.3 but has a sample complexity scaling as $\mathrm{poly}(\left(\frac{dH}{\varepsilon}\right)^A, B)$ and is computationally intractable. Our extension of Tensor-Plan naturally incorporates the expert demonstrations, while simultaneously (1) having low oracle requirements, (2) addressing the exponential sample complexity of TensorPlan, and (3) rendering the algorithm computationally efficient. Our approach is based only on finding value functions which satisfy the Bellman equation. Bellman error minimization approaches have appeared in other works (e.g. [JLM21; ZLKB20; CJ19]), but have always required a restrictive "Bellman closedness" assumption. As discussed, our lower bound construction is an extension of the recent remarkable lower bound of [WSG21], although several aspects of the construction have been modified to obtain better rates. In particular, we modified the reward/value functions, the feature mappings, and introduced an expert policy which differs from the optimal policy. For the proof, our setting is more complex as the learner has adaptive access to a second information source (the oracle), and a more sophisticated information-theoretic argument was needed to show that the oracle does not leak too much information.

## 6    Conclusion

We presented the DELPHI algorithm for RL with an interactive expert. We saw that, with $\tilde{\mathcal{O}}(d)$ oracle calls, exponential improvements in sample complexity are possible for RL with linearly-realizable optimal value functions. Compared to prior works on learning with an interactive expert, we also saw that DELPHI 's oracle requirements were smaller, and in fact are independent of the horizon of the MDP. It would be interesting and fruitful to resolve the gap between the oracle complexity required by DELPHI and the one obtained from our lower bound (either answer would be surprising to the authors). It would also be fruitful to study the case of linearly-realizable *action-value* functions in stochastic MDPs, which would potentially enable our method to be extended to the online setting.

## Acknowledgments and Disclosure of Funding

PA gratefully acknowledges funding from the Natural Sciences and Engineering Research Council of Canada (NSERC). Work done in part while PA was an intern at Amazon.

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
