# A Proof of Theorem 3.1

**Hyperparameters for the algorithm**   We define

$$E_d = 3d\frac{e}{e-1}\ln\{3 + 3\left(2B/\varepsilon\right)^2\} + 1 \tag{9}$$

$$n_{\texttt{rollout}} = \frac{2H^2(1+2B)^2 \log\left(\frac{2(E_d+1)}{\delta}\right)}{\varepsilon_{\texttt{target}}^2} \tag{10}$$

$$N = (E_d + 1)n_{\texttt{rollout}}HA \tag{11}$$

$$n_{\texttt{eval}} = \frac{50H^2(1+B^2)(d+1)\log\left(\frac{2(d+1)N}{\delta}\right)}{\varepsilon_{\texttt{target}}^2} \tag{12}$$

$$\varepsilon_{\texttt{eval}} = \sqrt{\frac{\log\left(\frac{2(d+1)N}{\delta}\right)}{2n_{\texttt{eval}}}} \tag{13}$$

$$\bar{\varepsilon}_{\texttt{eval}} = \sqrt{1 + B^2}\sqrt{d+1}\varepsilon_{\texttt{eval}} \tag{14}$$

$$\varepsilon_{\texttt{tol}} = 4\bar{\varepsilon}_{\texttt{eval}} \tag{15}$$

$$\varepsilon_{\texttt{roll}} = H(1+2B)\sqrt{\frac{\log(2(E_d+1)/\delta)}{2n_{\texttt{rollout}}}} \tag{16}$$

As we will see, $E_d$ is the upper bound on the number of parameters observed before global consistency holds, $n_{\texttt{eval}}$ is the number of samples taken to estimate each TD vector, $n_{\texttt{rollout}}$ is the number of rollouts executed for each parameter, $N$ is the maximum number of state-action pairs which will be seen by DELPHI , $\varepsilon_{\texttt{eval}}$ is the error between the estimated TD vector and the true TD vector, and $\varepsilon_{\texttt{roll}}$ is the error of replacing the expected total rewards with the average over several rollouts.

## A.1   Part 1: Concentration Inequalities

Recall that we write

$$\Delta_{s,a} = r(s,a) \oplus \left(\mathbb{E}_{P(s,a)}[\varphi(s')] - \varphi(s)\right)$$

for the true TD vector for state-action $(s,a)$. We write $\hat{\Delta}_{s,a}$ for any estimated TD vector resulting from Line 11, and $\tilde{\Delta}_{s,a}$ for any doubly-estimated TD vector resulting from Line 16.

**Lemma 3.2** (Concentration of $\hat{\Delta}_{s,a}$ (Line 11)). *For any $s, a \in \mathcal{S} \times \mathcal{A}$ that is observed throughout the execution* DELPHI *, with $n_{eval}$ samples in Line 11, we have that with probability $\geq 1 - \delta$, $\left\|\hat{\Delta}_{s,a} - \Delta_{s,a}\right\|_\infty \leq \varepsilon_{eval}$ and thus that $\langle 1 \oplus \theta, \hat{\Delta}_{s,a} - \Delta_{s,a}\rangle \leq \bar{\varepsilon}_{eval}$.*

*Proof.* Pick a single $s, a$, and omit the dependence on $s, a$ for cleanliness. Starting with the reward concentration: at every step we collect $n_{\texttt{eval}}$ iid samples from $R(s_i, a_i)$. By Hoeffding, since rewards are bounded in $[0, 1]$, the empirical average satisfies

$$\left|\frac{1}{n}\sum r_i - \mathbb{E}[r]\right| \overset{\text{w.p. } 1-\delta/((d+1)N)}{\leq} \sqrt{\frac{\log(2(d+1)N/\delta)}{2n_{\texttt{eval}}}}.$$

Onto the transition probabilities. Since the features are such that $\|\varphi\|_2 \leq 1$, we also have that $\|\varphi\|_\infty \leq 1$. Thus, each coordinate of the feature map is bounded by 1. By Hoeffding, for each coordinate $i$ of the vector, we have that

$$|\frac{1}{n}\sum_{j=1}^n \varphi_i^{(j)} - \mathbb{E}\varphi_i^{(j)}| \overset{\text{w.p. } 1-\delta/((d+1)N)}{\leq} \sqrt{\frac{\log(2(d+1)/\delta)}{2n_{\texttt{eval}}}}$$

Thus, with a union bound over these events for each coordinate $i$ and the event for the reward concentration, we have that w.p. $\geq 1 - \delta/N$

$$\left\|\hat{\Delta} - \Delta\right\|_\infty \leq \sqrt{\frac{\log\left(\frac{2(d+1)N}{\delta}\right)}{2n_{\texttt{eval}}}} = \varepsilon_{\texttt{eval}}.$$

Finally, by a union bound over all $N$ state-action pairs observed by DELPHI , we have that for every $s, a$ encountered:

$$\left\|\hat{\Delta}_{s,a} - \Delta_{s,a}\right\|_\infty \leq \sqrt{\frac{\log\left(\frac{2(d+1)N}{\delta}\right)}{2n_{\texttt{eval}}}} = \varepsilon_{\texttt{eval}} \text{ with probability } \geq 1 - \delta.$$

We further find that: $\left\|\hat{\Delta} - \Delta\right\|_2 \leq \sqrt{d+1}\varepsilon_{\texttt{eval}}$ (since $\Delta$ and $\hat{\Delta}$ are $(d+1)$-dimensional). By Cauchy-Schwartz this gives, $\forall \theta \in \texttt{Ball}_{\ell_2}(B)$:

$$|\langle 1 \oplus \theta, \hat{\Delta} - \Delta\rangle| \leq \|1 \oplus \theta\|_2 \left\|\hat{\Delta} - \Delta\right\|_2 = \sqrt{1 + \|\theta\|_2^2}\left\|\hat{\Delta} - \Delta\right\|_2 = \sqrt{1 + B^2}\sqrt{d+1}\varepsilon_{\texttt{eval}} = \bar{\varepsilon}_{\texttt{eval}}$$

$\square$

**Lemma 3.3** ($\tilde{\Delta}_{s,a}$ concentrates even more (Line 16)). *Similarly, for all $s, a$ where we call the oracle, with probability $1 - \delta$, we have $\left\|\tilde{\Delta}_{s,a} - \Delta_{s,a}\right\|_\infty \leq \varepsilon_{\texttt{eval}}/(2\sqrt{E_d})$, and thus, $\forall \theta \in \texttt{Ball}_{\ell_2}(B)$, $|\langle 1 \oplus \theta, \tilde{\Delta}_{s,a} - \Delta_{s,a}\rangle| \leq \bar{\varepsilon}_{\texttt{eval}}/(2\sqrt{E_d})$.*

*Proof.* Follows from the same proof as Lemma 3.2, just replace $n_{\texttt{eval}}$ by $4E_d n_{\texttt{eval}}$. $\square$

### A.2 Part 2: Optimism

**Lemma 3.4** ($\theta^\circ$ not eliminated). *With probability $\geq 1 - \delta$, $\theta^\circ \in \Theta_t$ for all iterations $t \in [E_d + 1]$.*

*Proof.* Let $\tilde{\Delta}_t$ be the TD vector which is added to $\Theta_t$ at time $t \in [E_d]$. Note that, by Lemma 3.3, for any given $t$, we have

$$|\langle 1 \oplus \theta^\circ, \tilde{\Delta}_t - \Delta_t\rangle| \leq \bar{\varepsilon}_{\texttt{eval}}/(2\sqrt{E_d}), \tag{17}$$

for all $t$, w.p. $\geq 1 - \delta$. The parameter set is defined as

$$\Theta_t = \{\theta : |\langle 1 \oplus \theta, \tilde{\Delta}_i\rangle| \leq \frac{\bar{\varepsilon}_{\texttt{eval}}}{2\sqrt{E_d}} \,\forall i \in [t]\}.$$

Note that, for $\theta^\circ$, this is equivalent to requiring

$$|\langle 1 \oplus \theta^\circ, \tilde{\Delta}_t - \Delta_t\rangle| \leq \frac{\bar{\varepsilon}_{\texttt{eval}}}{2\sqrt{E_d}},$$

since $\langle 1 \oplus \theta^\circ, \Delta_t\rangle = 0$ for all $\Delta_t$ (recall Equation 5). This holds by Eq. (17). $\square$

**Lemma 3.5** (Optimism). *Under the event of Lemma 3.4, we have $v_t(s_0) \geq v^\circ(s_0)$, $\forall t \in [E_d]$.*

*Proof (of Lemma 3.5).* Under the event of Lemma 3.4, $\theta^\circ$ is never eliminated from $\Theta_t$. Thus, by the optimistic update rule, we have $\theta_{t+1} = \arg\max_{\theta \in \Theta_t} \theta^\top \varphi(s_0)$, and since $\theta^\circ \in \Theta_t$, then

$$\theta_{t+1}^\top \varphi(s_0) = v_{\theta_{t+1}}(s_0) \geq (\theta^\circ)^\top \varphi(s_0) = v_{\theta^\circ}(s_0) = v^\circ(s_0)$$

. $\square$

### A.3 Part 3: Iteration Complexity

To bound the iteration complexity of our algorithm, we will need to introduce the notion of *Eluder* dimension. We use a simplified form introduced by [LKFS21], although the first definition is due to [RVR13].

**Definition A.1** (Eluder dimension [LKFS21]). *Let $\mathcal{F}$ be a real-valued function class on domain $\mathcal{X}$. Fix a reference function $f^\star \in \mathcal{F}$, and a scale $\varepsilon$. The **Eluder dimension** of $\mathcal{F}$ at a scale $\varepsilon$, w.r.t. $f^\star$, is the length $\tau \in \mathbb{N}$ of the longest sequence of points $((x_1, f_1), \dots (x_\tau, f_\tau))$ such that*

$$\forall i \in [\tau]: \quad |f_i(x_i) - f^\star(x_i)| > \varepsilon, \quad and \quad \sum_{j<i}(f_i(x_j) - f^\star(x_j))^2 \leq \varepsilon^2. \tag{18}$$

*An Eluder sequence of length $\tau$ (with respect to $f^\star$) is any sequence $(x_i, f_i)_{i=1}^\tau$ which satisfies Eq (18) for each $i$.*

In other words, the Eluder dimension is the length of the longest sequence of points such that, for each $i$, we can find a new function $f_i$ which is large with respect to $f^\star$ on $x_i$ but correctly fits $f^\star$ on historical data $x_1, \ldots, x_j$, for $j < i$. We will use the folllowing bound for the Eluder dimension of linear functions with $d$-dimensional parameters.

**Lemma A.2** ([RVR13]). *For any $f^\star$, the Eluder dimension of the function class $\mathcal{F} = \{f_\theta(x) = \theta^\top x\}$, assuming that $\|\theta\|_2 \leq B$ and $\|x\|_2 \leq \gamma$, is*

$$dim_E(\mathcal{F}, \varepsilon) \leq 3d \frac{e}{e-1} \ln\{3 + 3(2B/\varepsilon)^2\} + 1 = \mathcal{O}(d\ln(B/\varepsilon)).$$

This gives us enough to prove our iteration bound – we will show that the sequence of linear parameters chosen by our algorithm together with each new TD vector obtained from the oracle forms an Eluder sequence with respect to $\theta^\star$

**Lemma 3.6** (Iteration Complexity). *With probability $\geq 1 - 2\delta$, the iteration complexity of the algorithm is at most the Eluder dimension at scale $\bar{\varepsilon}_{\mathtt{eval}}$, i.e. $E_d = \mathcal{O}(d\ln(B/\bar{\varepsilon}_{\mathtt{eval}}))$.*

*Proof (of Lemma 3.6).* Assume the events of Lemma 3.2 and Lemma 3.3, which happen together with probability $\geq 1 - 2\delta$. Our function class is $\mathcal{F} = \{f_\theta(x) = \langle 1 \oplus \theta, x \rangle\}$. This is a subset of all linear functions on $d + 1$ dimensions, so by Lemma A.2,[7] it has Eluder dimension at least $\mathcal{O}((d+1)\ln(B/\varepsilon)) = \mathcal{O}(d\ln(B/\varepsilon))$. We pick $f^\star = f_{\theta^\circ}$. We will show that $\forall i$, the sequence $(\tilde{\Delta}_i, f_{\theta_i})$ forms an Eluder sequence with respect to $f^\star$, from which it will follow that its length is bounded by $E_d$. We do this by induction. The base case is obvious. By the constraint set definition (Line 17), we have $|f_{\theta_i}(\tilde{\Delta}_j)| = |\langle 1 \oplus \theta_i, \tilde{\Delta}_j \rangle| \leq \frac{\bar{\varepsilon}_{\mathtt{eval}}}{2\sqrt{E_d}} \forall j < i$. Assuming the event of Lemma 3.4, we also have $|f_{\theta^\circ}(\tilde{\Delta}_j)| \leq \frac{\bar{\varepsilon}_{\mathtt{eval}}}{2\sqrt{E_d}}$, since $\theta^\circ \in \Theta_{i-1}$. Thus

$$|f_{\theta_i}(\tilde{\Delta}_j) - f_{\theta^\circ}(\tilde{\Delta}_j)| \leq 2\frac{\bar{\varepsilon}_{\mathtt{eval}}}{2\sqrt{E_d}} \forall i \implies \sum_{j<i}(f_{\theta_i}(\tilde{\Delta}_j) - f_{\theta^\circ}(\tilde{\Delta}_j))^2 \leq \bar{\varepsilon}_{\mathtt{eval}}^2 \frac{i-1}{E_d} \leq \bar{\varepsilon}_{\mathtt{eval}}^2,$$

where in the last inequality we have used $i - 1 \leq E_d$ by the induction hypothesis. Thus, the second condition of the Eluder dimension is satisfied. For the first condition, we want to show that $|f_{\theta_i}(\tilde{\Delta}_i) - f_{\theta^\circ}(\tilde{\Delta}_i)| > \bar{\varepsilon}_{\mathtt{eval}}$. Note that, by Lemma 3.4, we have $|f_{\theta^\circ}(\tilde{\Delta}_i)| \leq \bar{\varepsilon}_{\mathtt{eval}}/(2\sqrt{E_d}) \leq \bar{\varepsilon}_{\mathtt{eval}}$. Recall from Line 13 that $\left|\langle \hat{\Delta}_i, 1 \oplus \theta_i \rangle\right| = |f_{\theta_i}(\hat{\Delta}_i)| > \varepsilon_{\mathtt{tol}}$. Using concentration and linearity, we have $|f_{\theta_i}(\hat{\Delta}_i)| - |f_{\theta_i}(\tilde{\Delta}_i)| \leq 2\bar{\varepsilon}_{\mathtt{eval}}$. Thus, $|f_{\theta_i}(\tilde{\Delta}_i)| \geq \varepsilon_{\mathtt{tol}} - 2\bar{\varepsilon}_{\mathtt{eval}} = 2\bar{\varepsilon}_{\mathtt{eval}}$ since $\varepsilon_{\mathtt{tol}} = 4\bar{\varepsilon}_{\mathtt{eval}}$. Putting this together gives

$$|f_{\theta_i}(\tilde{\Delta}_i) - f_{\theta^\circ}(\tilde{\Delta}_i)| \geq f_{\theta_i}(\tilde{\Delta}_i) - f_{\theta^\circ}(\tilde{\Delta}_i) \geq 2\bar{\varepsilon}_{\mathtt{eval}} - \bar{\varepsilon}_{\mathtt{eval}} = \bar{\varepsilon}_{\mathtt{eval}}.$$

And we are done. $\qquad\square$

### A.4 Part 4: Consistency, and Putting Everything Together

**Lemma 3.7** (Consistency $\implies$ accurate prediction). *If $m$ rollouts have occured without any inconsistencies (i.e., the if statement of Line 13 never gets triggered), then $v^{\pi_\theta}(s_0) > v_\theta(s_0) - 5H\bar{\varepsilon}_{\mathtt{eval}} - \varepsilon_{\mathtt{roll}}$ with probability $\geq 1 - 3\delta$.*

*Proof (of Lemma 3.7).* Assume the event of Lemmas 3.2 and Lemmas 3.3, which together happen with probability $\geq 1 - 2\delta$. The third event which we assume will be introduced shortly.

For cleanliness, let us write $\theta$ for the final parameter which observes $m$ rollouts without consistency break. Observe the following calculation:

---

[7](and using that $\mathcal{F} \subseteq \mathcal{F}' \implies dim_E(\mathcal{F}, \varepsilon) \leq dim_E(\mathcal{F}', \varepsilon)$)

$$v^{\pi_\theta}(s_0) = \mathbb{E}_{\pi_\theta}\Big[\sum_{j\in[H]} r_{S_j,A_j}\Big]$$

$$= \mathbb{E}\left[\left\langle\left(\sum_{j\in[H]} r_{S_j,A_j}\right)\oplus\varphi(s_{H+1}), 1\oplus\theta\right\rangle\right] \qquad (\varphi(s_{H+1})=0)$$

$$= \mathbb{E}\left[\langle\varphi(s_0),\theta\rangle + \sum_{j\in[H]}\langle r_{S_j,A_j}\oplus(\varphi(S_{j+1})-\varphi(S_j)), 1\oplus\theta\rangle\right] \qquad (\text{telescoping sum})$$

$$= \langle\varphi(s_0),\theta\rangle + \mathbb{E}\left[\sum_{j\in[H]}\langle r_{S_j,A_j}\oplus\big(P_{S_j,A_j}\varphi(\cdot)-\varphi(S_j)\big), 1\oplus\theta\rangle\right],$$

Now observe that after $n_{\texttt{rollout}}$ number of rollouts, we have $n_{\texttt{rollout}}$ unbiased estimates of the expected trajectories. Thus we can approximate $\mathbb{E}\left[\sum_{j\in[H]}\langle r_{S_j,A_j}\oplus\big(P_{S_j,A_j}\varphi(\cdot)-\varphi(S_j)\big), 1\oplus\theta\rangle\right] \approx \frac{1}{n_{\texttt{rollout}}}\sum_{i=1}^{n_{\texttt{rollout}}}\mathbb{E}[r_{S_j^i,A_j^i}] \oplus \big(P_{S_j^i,A_j^i}\varphi(\cdot)-\varphi(S_j^i)\big)$, where $S_j^i$ and $A_j^i$ are the states and actions in horizon $j\in[H]$ of rollout $i\in[n_{\texttt{rollout}}]$. More precisely, using Hoeffding's and that $\langle\mathbb{E}[r_{S_j^i,A_j^i}]\oplus\big(P_{S_j^i,A_j^i}\varphi(\cdot)-\varphi(S_j)\big)], 1\oplus\theta\rangle \leq 1+2B$, we can get

$$\left|\frac{1}{n_{\texttt{rollout}}}\sum_{i\in[m]}\left(\sum_{j\in[H]}\langle\Delta_j^i, 1\oplus\theta\rangle\right) - \left(\mathbb{E}\sum_{j\in[H]}\langle\Delta_j^i, 1\oplus\theta\rangle\right)\right| \leq H(1+2B)\sqrt{\frac{\log(2(E_d+1)/\delta)}{2m}} := \varepsilon_{\texttt{roll}},$$

w.p. $\geq 1-\frac{\delta}{E_d+1}$, where we wrote $\Delta_j^i = \mathbb{E}[r_{S_j^i,A_j^i}]\oplus\big(P_{S_j^i,A_j^i}\varphi(\cdot)-\varphi(S_j)\big)$. By a union bound this happens for all $t\in[E_d+1]$ with probability $\geq 1-\delta$. Picking up where we left off:

$$v^{\pi_\theta}(s_0) \geq \langle\varphi(s_0),\theta\rangle + \frac{1}{n_{\texttt{rollout}}}\sum_{i=1}^{n_{\texttt{rollout}}}\sum_{j\in[H]}\langle\mathbb{E}[r_{S_j^i,A_j^i}]\oplus\big(P_{S_j^i,A_j^i}\varphi(\cdot)-\varphi(S_j)\big)], 1\oplus\theta\rangle - \varepsilon_{\texttt{roll}}$$

$$\geq \langle\varphi(s_0),\theta\rangle + \frac{1}{m}\sum_{i=1}^{m}\sum_{j\in[H]}\langle\hat{\Delta}_{s_j^i,a_j^i}, 1\oplus\theta\rangle - H\bar\varepsilon_{\texttt{eval}} - \varepsilon_{\texttt{roll}} \qquad (\text{evaluation error})$$

$$\geq v_\theta(s_0) - \frac{1}{m}\sum_{i=1}^{m}\sum_{j\in[H]} 4\bar\varepsilon_{\texttt{eval}} - H\bar\varepsilon_{\texttt{eval}} - \varepsilon_{\texttt{roll}} \qquad (\text{consistency holds})$$

$$= v_\theta(s_0) - 5H\bar\varepsilon_{\texttt{eval}} - \varepsilon_{\texttt{roll}}$$

and we are done. The second inequality follows from Lemma 3.2, which holds for all steps in all $m$ trajectories in all $E_d+1$ iterations. The third inequality holds since, for every $S_j^i$, there exists a consistent action, i.e. an action such that $|\langle\hat{\Delta}_j^i, 1\oplus\theta\rangle| \leq \varepsilon_{\texttt{tol}} = 4\bar\varepsilon_{\texttt{eval}}$. $\qquad\square$

### A.5 Simulator inaccuracy and misspecification

We start with the case of inaccurate simulators. Recall that we say that a simulator is $\lambda$-*innacurate* if the samples obtained are of the form $(\Pi(r+\lambda_{s,a}), s')$, for any $(s,a)$ and for some constant $\lambda_{s,a}$ that satisfies $|\lambda_{s,a}|\leq\lambda$.

**Theorem 3.8** (DELPHI with misspecification). *Redefine $n'_{eval}=4n_{eval}$ (Eq. 12) and all subsequent hyperparameters which depend on $n'_{eval}$. Then we have that, for all MDPs that are at most $\frac{\bar\varepsilon_{eval}}{8\sqrt{E_d}}$-misspecified or for all simulators that are at most $\frac{\bar\varepsilon_{eval}}{4\sqrt{E_d}}$-innacurate, the conclusions of Theorem 3.1 continue to hold when running DELPHI with the new hyperparameters.*

We can repeat the proof of DELPHI, and in fact the only difference will be in Part 1 of the proof (Lemmas 3.2 and 3.3) will continue to hold, which also implies that the rest of the proof will continue to hold.

**Lemma A.3** (Concentration of $\hat{\Delta}_{s,a}$ (Line 11)). *For any $s, a \in \mathcal{S} \times \mathcal{A}$ that is observed throughout the execution* DELPHI *, with $n_{\text{eval}}$ samples in Line 11, we have that with probability $\geq 1 - \delta$, $\left\| \hat{\Delta}_{s,a} - \Delta_{s,a} \right\|_\infty \leq \varepsilon_{\text{eval}}$ and thus that $\langle 1 \oplus \theta, \hat{\Delta}_{s,a} - \Delta_{s,a} \rangle \leq \bar{\varepsilon}_{\text{eval}}$.*

**Lemma A.4** ($\tilde{\Delta}_{s,a}$ concentrates even more (Line 16)). *Similarly, for all $s, a$ where we call the oracle, with probability $1 - \delta$, we have $\left\| \tilde{\Delta}_{s,a} - \Delta_{s,a} \right\|_\infty \leq \varepsilon_{\text{eval}}/(2\sqrt{E_d})$, and thus, $\forall \theta \in \text{Ball}_{\ell_2}(B)$, $|\langle 1 \oplus \theta, \tilde{\Delta}_{s,a} - \Delta_{s,a} \rangle| \leq \bar{\varepsilon}_{\text{eval}}/(2\sqrt{E_d})$.*

*Proof (of Lemmas A.3 and A.4.* We repeat the proof of Lemmas 3.2 and 3.3. Note that $\hat{\Delta}_{s,a}$ is an average of i.i.d. random variables with mean $\Delta'_{s,a} = (r(s,a) + \lambda_{s,a}) \oplus P_{s,a}\varphi - \varphi(s)$. As before, by Hoeffdings, we have $|\langle 1 \oplus \theta, \hat{\Delta} - \Delta' \rangle| \leq \bar{\varepsilon}_{\text{eval}}/2$, where the factor of $1/2$ is due to the definition of $n'_{\text{eval}}$. This gives that

$$|\langle 1 \oplus \theta, \hat{\Delta} - \Delta \rangle| \leq |\langle 1 \oplus \theta, \hat{\Delta} - \Delta' \rangle| + |\langle 1 \oplus \theta, \Delta' - \Delta \rangle| \leq \bar{\varepsilon}_{\text{eval}}/2 + \lambda \leq \bar{\varepsilon}_{\text{eval}}/2 + \bar{\varepsilon}_{\text{eval}}/4\sqrt{E_d} \leq \bar{\varepsilon}_{\text{eval}}$$

The proof of the second part follows similarly, since we have

$$|\langle 1 \oplus \theta, \tilde{\Delta} - \Delta \rangle| \leq |\langle 1 \oplus \theta, \tilde{\Delta} - \Delta' \rangle| + |\langle 1 \oplus \theta, \Delta' - \Delta \rangle| \leq \bar{\varepsilon}_{\text{eval}}/4\sqrt{E_d} + \lambda \leq \bar{\varepsilon}_{\text{eval}}/2\sqrt{E_d}$$

$\square$

Next we handle the value misspecification case. Recall that the MDP is $\eta$-*misspecified* for the expert policy $\pi^\circ$ and the features $\varphi$ if there exists $\theta^\circ$ such that $\sup_s |v^\circ(s) - \langle \varphi(s), \theta^\circ \rangle| \leq \eta$. Here, we use a reduction argument to show that an $\eta$-misspecified MDP can be reduced to a $2\eta$-innacurate simulator of a realizable MDP. Namely, using the same reduction as Appendix D of [WAJAYJS21], we can construct an alternative MDP $\mathcal{M}'$ such that $\mathcal{M}'$ is realizable but is a $2\eta$-inaccurate simulator of $\mathcal{M}$. The result then follows from the first part of the proof.

# B  Proof of Section 4

## B.1  The MDP construction

Theorems 4.1 and 4.3 use the same MDP construction, which is inspired by the recent and remarkable lower bound of [WSG21] (itself an extension of the lower bound of [WAS21]). We give an overview of the MDP and its features, and describe the changes from the original construction of [WSG21]. The state space consists of a hypercube in $p$ dimensions, $\mathcal{S} = \{\pm 1\}^p$, for some $p \in \mathbb{N}$, and assume for simplicity that $p$ is divisible by 4. More specifically, the state space at stage $h$ also contains the history of vectors encountered so far $(s_1, \ldots, s_h)$, which is uniquely defined as the transitions are deterministic. The dimension $p$ will end up being $p \approx \sqrt{d}$ for the value-based lower bound and $p \approx d$ for the policy-based lower bound. We write $\rho(\cdot, \cdot)$ for the *Hamming* distance on $\mathcal{S}$, recalling that is a bilinear function of its arguments, i.e. $\rho(s_1, s_2) = \frac{1}{2}(p - \langle s_1, s_2 \rangle)$. The action set is $\mathcal{A} = [p]$, and each action $a \in [p]$ will correspond to flipping the $a^{\text{th}}$ bit of the current state. Each trajectory of horizon $H$ has $K$ *phases* ($K \in \mathbb{N}$), and each phase consists of $p$ steps (thus, $H = Kp$). We write $s_0, s_1, \ldots, s_k, \ldots s_K$ for the states reached at the end of each phase, and when we need to we will use $s_{k,i}$ for a state at step $i$ of phase $k$. The start state is $s_0 = \vec{1}$, the all-ones vector.

There is a special "goal state" $s^\star$, and reward is given only if a) at the end of any phase, $\rho(s_k, s^\star) \leq p/4$, or b) the learner reaches horizon $H$, i.e. the end of phase $K$. The reward function decays geometrically, and is defined according to the sequence of states $s_0, \ldots s_k$ obtained at the end of each phase. Letting $g(s_1, s_2) := 1 - \rho(s_1, s_2)/p$ denote one minus the proportion of bits where two states differ, the reward function for reaching a $p/4$-neighbourhood of $s^\star$ at stage $k$ (condition a) just described) is deterministic and has value

$$r_{w^\star}((s_i)_{i=1}^k) = \left( \prod_{i=1}^k g(s_{i-1}, s_i) \right) g(s_i, s^\star) \tag{19}$$

The reward function at stage $H$ (condition b) above) is always given and has the same expectation as Eq. (19), but will be a Bernoulli random variable.

Modulo some exceptions (to be described shortly), the transition function is deterministic and is defined via $\mathcal{P}(\tau(s, a)|s, a) = 1$, where $\tau(s, a) = (s_1, \ldots, -s_a, \ldots s_p)$ corresponds to the new vector obtained from flipping the bit at index $a \in [p]$. The exceptions to these transition dynamics are if 1) a state within a $p/4$-neighbourhood of $s^\star$ is reached, or 2) a move is repeated. In the first case, the MDP transitions to a game-over state $\perp$ (after which nothing else is possible and no reward is given). In the second case, if the move is repeated within the first $p/4$ steps of a phase then the MDP similarly transitions to $\perp$, and if the move is repeated in the second $3p/4$ turns then the current state becomes frozen until the end of the current phase (i.e. no further bit flips are allowed). This implies that each bit can only be flipped once in each phase, and further that $g(s_{i-1}, s_i) \leq 3/4$ for each "legal" trajectory in the MDP (thus, the reward decays geometrically, cf. Equation (19)).

So far, the main modification in our construction from that of [WSG21] is the reward function. Their $g$ function is chosen to be $2^{\text{nd}}$ order in $\rho(s_1, s_2)$, while ours is linea in $\rho$. We now introduce the definition of the expert policy: we simply choose it to be the one which will flip the *earliest* index such that $s_{k,i}$ differs from $s^\star$ and such that its index has not yet been played in the round. If no such index exists, or if $s_{k,i} = s^\star$, the expert policy will simply freeze the current round by repeating the earliest index that has already been played. (Note that there is always a repeated index in this case, and that repeating an index will lead to freezing instead of termination, since at the start of any phase $k$ the trajectory must satisfy $\rho(s_k, s^\star) > p/4$, otherwise the trajectory would have terminated). It will turn out that the expert's trajectory *from the start state* will be identical to that of the optimal policy (and, thus, will be equally difficult to compete with, in the sense of Eq. (2)). While our expert policy might seem optimal, this need not be the case for arbitrary states since it might sometimes be more advantageous for a state in phase $k$ which can no longer reach the $p/4$-neighbourhood of $s^\star$ to simply aim at minimizing the inevitable factor of $g(s_{k-1}, s_k)$ that it will incur. The following lemma (proved in Appendix B.2) gives an expression for the value function corresponding to our expert policy, and shows that the value function satisfies Assumption 2.2.

**Lemma B.1.** *Let $s_{k,i}$ be a state in round $i$ of phase $k$. Note that, from $s_{k,i}$, the state $s_{k+1}$ that $\pi^\circ$ will reach at the end of the current phase is deterministic. The value function of $\pi^\circ$ is*

$$v^\circ(s_{k,i}) = \left( \prod_{k' \in [k]} g(s_{k'-1}, s_{k'}) \right) g(s_k, s_{k+1}) g(s_{k+1}, s^\star),$$

*and furthermore this is linear with features $\varphi_v$ of dimension $d = \Theta(p^2)$.*

The next lemma gives an expression showing that this same expert policy is linear in a different set of features $\varphi_\pi$, and thus that the expert policy satisfies Assumption 4.2.

**Lemma B.2.** *There exists a feature map $\varphi_\pi$ and parameter $\theta_\pi$ of dimension $d = \Theta(p)$ such that*

$$\pi^\circ(s_{k,i}) = \mathrm{argmax}_a\{\langle \varphi_\pi(s, a), \theta_\pi \rangle\}$$

This lemma is also proved in Appendix B.2.

The essence of our lower bound is that each oracle call will reveal one bit of the secret state $s^\star$, and thus without $\tilde{\Omega}(p)$ calls and learner will need exponentially-many exploratory samples to solve this MDP. Thus, for the value-based lower bound, an agent with a $d$-dimensional feature map must be given an MDP which has a $p = \Theta(\sqrt{d})$-dimensional state space. However, for the policy-based lower bound, we can give the agent a $p = \Theta(d)$-dimensional MDP.

**Information-theoretic lower bound** Information-theoretically, a learning algorithm can solve this MDP only if they recover the secret state $s^\star$. We let $\mathrm{MDP}(p, K)$ denote an instance of the above MDP with dimension $p$ and parameters $K$. Following the approach of [WSG21], we prove the sample complexity hardness by reducing each $\mathrm{MDP}(p, K)$ to an instance of an abstract game called $\mathrm{CUBEGAME}(p, K)$. Details on $\mathrm{CUBEGAME}$, and the proof of the following theorem, are deferred to Appendix B.3. The main result is the following:

**Theorem B.3.** *Any learner which solves $\mathrm{MDP}(p, K)$ can be used to solve $\mathrm{CUBEGAME}(p, K)$. Unless the number of oracle calls is $\Omega(\frac{p}{\log p})$, any learner which is $0.01$-optimal on $\mathrm{CUBEGAME}(p, K)$, with a sample complexity of $N$, will need*

$$N \geq 2^{\Omega(p \wedge K)}.$$

Combined with the fact that a learner provided with $d$-dimensional features is given an MDP with parameter $p \approx \sqrt{d}$, this gives the result of Theorem 4.1.

## B.2 Proofs of Lemma B.1, B.2

We need some more notation in order to linearize the value function. Let $s_{k,i} \neq \perp$ be a state in step $i$ of phase $k$. We define the variable $\mathrm{ct}_{k,i}^{\mathrm{flip}} = \rho(s_{k,0}, s_{k,i})$, which simply measures the number of components flipped so far in round $k$ of step $i$, and $\mathrm{fix}_{k,i} \in \{0, 1\}^p$ which is a vector with 1 at a component $j$ if said component is currently frozen (i.e. if it has been played or if the entire state has been frozen), and 0 otherwise. Similarly, there are two scalars $e_{k,i}^{\mathrm{fix}}$ and $e_{k,i}^{\neg\mathrm{fix}}$ which simply count the number of components which disagree with $s^\star$ that are currently frozen (for $e_{k,i}^{\mathrm{fix}}$) or not frozen (for $e_{k,i}^{\neg\mathrm{fix}}$). Note that only $e_{k,i}^{\mathrm{fix}}$ and $e_{k,i}^{\neg\mathrm{fix}}$ depend on $s^\star$, and in fact we have:

$$e_{k,i}^{\mathrm{fix}} = \frac{1}{2}(\langle \vec{1}, \mathrm{fix}_{k,i} \rangle - \langle \mathrm{fix}_{k,i} \cdot s_{k,i}, s^\star \rangle) \tag{20}$$

$$e_{k,i}^{\neg\mathrm{fix}} = \frac{1}{2}(\langle \vec{1}, \neg\mathrm{fix}_{k,i} \rangle - \langle \neg\mathrm{fix}_{k,i} \cdot s_{k,i}, s^\star \rangle), \tag{21}$$

where $\mathrm{fix}_{k,i} \cdot s_{k,i}$ denotes component-wise multiplication, i.e. $(\mathrm{fix}_{k,i} \cdot s_{k,i})_j = (\mathrm{fix}_{k,i})_j \cdot (s_{k,i})_j$. Now, we have that:

**Lemma B.4** (Value of $v^\circ$). *Let $s_{k,i}$ be a state in round $i$ of phase $k$. Let $s_{k+1}$ denote the state that $\pi^\circ$ will reach at the end of the current phase when starting from $s_{k,i}$ (and note that this choice is deterministic given $s_{k,i}$, and that it may not be in the $p/4$-neighbourhood of $s^\star$). Then we have:*

$$v^\circ(s_{k,i}) = \left( \prod_{k' \in [k]} g(s_{k'-1}, s_{k'}) \right) g(s_k, s_{k+1}) g(s_{k+1}, s^\star),$$

*or, overloading notation and letting $g(x) = 1 - x/p$, we have*

$$v^\circ(s_{k,i}) = \left( \prod_{k' \in [k]} g(s_{k'-1}, s_{k'}) \right) g(ct_{k,i}^{\text{flip}} + e_{k,i}^{\neg \text{fix}}) g(e_{k,i}^{\text{fix}}). \tag{22}$$

*Proof.* Identical to [WSG21, Lemma 4.9]   □

**Lemma B.5.** *The value function $v^\circ$ is linear in features $\varphi_v$ with dimension $d = \Theta(p^2)$.*

*Proof.* Starting from Equation 22, we observe that only $e_{k,i}^{\text{fix}}$ and $e_{k,i}^{\neg \text{fix}}$ depend on $s^\star$. The first term in parentheses in simply a scalar which multiplies the features. We thus calculate linear expressions for $x = ct_{k,i}^{\text{flip}} + e_{k,i}^{\neg \text{fix}}$ and $y = e_{k,i}^{\text{fix}}$. Starting with $y = e_{k,i}^{\text{fix}}$, we have:

$$y = \frac{1}{2} \left( \langle \vec{1}, \text{fix}_{k,i} \rangle - \langle \text{fix}_{k,i} \cdot s_{k,i}, s^\star \rangle \right) = a + \langle b, s^\star \rangle,$$

where $a = \frac{1}{2} \langle \vec{1}, \text{fix}_{k,i} \rangle$ and $b = -\frac{1}{2} \text{fix}_{k,i} \cdot s_{k,i}$. Similarly:

$$x = ct_{k,i}^{\text{flip}} + \frac{1}{2} \left( \langle \vec{1}, \neg \text{fix}_{k,i} \rangle - \langle \neg \text{fix}_{k,i} \cdot s_{k,i}, w^\star \rangle \right) = c + \langle d, s^\star \rangle,$$

where $c = ct_{k,i}^{\text{flip}} + \frac{1}{2}(\langle \vec{1}, \neg \text{fix}_{k,i} \rangle)$ and $d = -\frac{1}{2} \neg \text{fix}_{k,i} \cdot s_{k,i}$. Thus we have: $g(y) = 1 - y/p = 1 - (a + \langle d, s^\star \rangle)/p = (1 - a/p) - \langle \bar{b}, \bar{s}^\star \rangle = a' + \langle \bar{b}, \bar{s}^\star \rangle$, where $a' = (1 - a/p)$, $\bar{b} = -b/\sqrt{p}$ and $\bar{s}^\star = s^\star/\sqrt{p}$. Similarly, $g(x) = 1 - x/p = 1 - (c + \langle d, s^\star \rangle)/p = c' - \langle \bar{d}, \bar{s}^\star \rangle$, where $c' = 1 - c/p$ and $\bar{d} = d/\sqrt{p}$. Putting this together we have that

$$v^\circ(s_{k,i}) = \left( \prod_{k' \in [k]} g(s_{k'-1}, s_{k'}) \right) g(x) g(y)$$

$$= \left( \prod_{k' \in [k]} g(s_{k'-1}, s_{k'}) \right) (a' + \langle \bar{b}, \bar{s}^\star \rangle)(c' + \langle \bar{d}, \bar{s}^\star \rangle)$$

$$= \left( \prod_{k' \in [k]} g(s_{k'-1}, s_{k'}) \right) \left( a'c' + c'\langle \bar{b}, \bar{s}^\star \rangle + a'\langle \bar{d}, \bar{s}^\star \rangle + \langle \bar{b}, \bar{s}^\star \rangle \langle \bar{d}, \bar{s}^\star \rangle \right)$$

$$= \left( \prod_{k' \in [k]} g(s_{k'-1}, s_{k'}) \right) \left( a'c' + \langle c'\bar{b} + a'\bar{d}, \bar{s}^\star \rangle + \langle \bar{b} \otimes \bar{d}, \bar{s}^\star \otimes \bar{s}^\star \rangle \right),$$

where we have use a property of the tensor product that $\langle a_1, b_1 \rangle \langle a_2, b_2 \rangle = \langle a_1 \otimes a_2, b_1 \otimes b_2 \rangle$, where $a_1 \otimes a_2, b_1 \otimes b_2 \in \mathbb{R}^{p \times p}$ and their inner product is interpreted as the inner product between the vectorized matrices. Thus, if we take

$$\theta_v = 1 \oplus \bar{s}^\star \oplus (\bar{s}^\star \otimes \bar{s}^\star) \in \mathbb{R}^{1 + p + p^2}$$

and

$$\varphi(s_{k,i}) = \left( \prod_{k' \in [k]} g(s_{k'-1}, s_{k'}) \right) \left( a'c' \oplus (c'\bar{b} + a'\bar{d}) \oplus (\bar{b} \otimes \bar{d}) \right) \in \mathbb{R}^{1 + p + p^2},$$

then we have $v^\circ(s_{k,i}) = \langle \varphi(s_{k,i}), \theta_v \rangle$ as desired. Thus $v^\circ$ is linear with features in dimension $1 + p + p^2$. Note that the norm of the features and the parameter $\theta_v$ is also bounded by constants.   □

This completes the proof for $v^\circ$-linearity. We next tackle the analogous statement for $\pi^\circ$-linearity.

**Lemma B.6.** *There exists a feature map $\varphi_\pi$ and parameter $\theta_\pi$ of dimension $d \approx p$ such that*

$$\pi^\circ(s_{k,i}) = \text{argmax}_a \{ \langle \varphi_\pi(s, a), \theta_\pi \rangle \}$$

*Proof.* Let $s_{k,i}$ be a state of interest. Recall that $\pi^\circ$ will either 1) flip the earliest index which has not been fixed such that the value of $s_{k,i}$ at that index disagrees with $s^\star$, or 2) if no such index exists, freeze the current round by playing a frozen index. First consider a state $s_{k,i}$ such that $\pi^\circ(s_{k,i})$ will flip an index. Since the index flipped was previously incorrect and will thereafter agree with $s^\star$ on that bit, this corresponds to minimizing the distance between $s'$ and $s^\star$ amongst all possible $s'$ which can be reached in one step from $s_{k,i}$ (i.e. amongst all possible $\{\tau(s_{k,i}, a)\}_a$, recalling that $\tau(s_{k,i}, a)$ is our notation for the transition function of the MDP). Thus, $\pi^\circ(s_{k,i}) \in \mathrm{argmin}_a\{\rho(\tau(s_{k,i}, a), s^\star)\}$. This can be written linearly as $\mathrm{argmin}_a\{\frac{1}{2}(p - \langle\tau(s_{k,i}, a), s^\star\rangle)\} = \mathrm{argmax}_a\{\frac{1}{2}(\langle\tau(s_{k,i}, a), s^\star\rangle - p)\} = \mathrm{argmax}_a\{\frac{1}{2}(\langle p \oplus \tau(s_{k,i}, a), s^\star\rangle, 1 \oplus s^\star\rangle)\}$. The second case is that $\pi^\circ$ will freeze the round at the state $s_{k,i}$. This means that there are no indices which are incorrect that have not been frozen in this round. Again, this corresponds to minimizing the distance between $\tau(s_{k,i}, a)$ and $s^\star$: all other choices will either result in a game over (which has 0 value) or will flip an incorrect bit (which increases the distance). Thus, again we have $\pi^\circ(s_{k,i}) \in \mathrm{argmin}_a\{\rho(\tau(s_{k,i}, a), s^\star)\} = \mathrm{argmax}_a\{\frac{1}{2}(\langle p \oplus \tau(s_{k,i}, a), s^\star\rangle, 1 \oplus s^\star\rangle)\}$.

Thus, in either case, we have that $\pi^\circ$ is linear with features $\varphi_\pi(s, a) = p \oplus \tau(s_{k,i}, a)$ and $\theta^\circ = 1 \oplus s^\star$. Since the definition of the $\mathrm{argmax}$ is scale-insensitive, we can further normalize to obtain that the features are bounded in magnitude by a constant. $\qquad\square$

## B.3 CUBEGAME with expert advice, and Proof of Theorem B.3

Following the approach of [WSG21], we give our lower bound by providing a reduction to an abstract game called CUBEGAME. Any learning algorithm which can solve the MDPs in our construction can also be used to solve CUBEGAME, and thus it follows that the learner will be subject to the same lower bound. For our setting, we modify the reward function CUBEGAME and augment the learner with the ability to query an expert, which will behave identically to the expert policy which we have defined in our MDPs (that is, it will flip the first bit which is incorrect or give a special actions to indicate if all of the bits are correct). For the rest of this section, when referring to CUBEGAME we are referring to our modified version.

In [WSG21] it is shown that any algorithm which outputs a $0.01-$optimal answer for the (expert-less) CUBEGAME will need a query complexity of $N \geq 2^{\Omega(p \wedge K)}$. In what follows, we will provide the analogous proof of this for our modified game. Thus, the main result of this section is the following, which states that if the learner is not given a budget of $\Omega(p/\log p)$ expert queries, the sample complexity remains exponential. A learning algorithm for CUBEGAME will be called a *planner*, and a planner which returns a 0.01-optimal answer at the end of CUBEGAME will be called *sound*.

**Rules of CUBEGAME**  CUBEGAME is a bandit-like game which is defined by two parameters: a length $K \in \mathbb{N}_+$ and a dimension $p \in \mathbb{N}_+$. The "action space" is $W = \{\pm 1\}^p$. Recall that $\rho(x, y) = \frac{1}{2}(p - x^\top y)$ is the Hamming distance between two vectors in $W$. The secret parameter which solves the game is housed in the set $W^\star = \{w \in W \mid \frac{p}{4} \leq \rho(\vec{1}, w) \leq \frac{3p}{4}\}$. The planner can only input sequences of vectors where each vector is sufficiently far from the previous one. Formally, for any $k \in [K]$, we let $W^{\circ k} = \{(w_i)_{i \in [k]} \in W^k \mid \forall i \in [k] : \rho(w_{i-1}, w_i) \geq p/4\}$, with $w_0 := \vec{1}$. The action space is: $\mathcal{A} = \cup_{k \in [K]} W^{\circ k}$, thus the planner can input any sequence of length $k \leq K$ satisfying that $(w_i) \in W^{\circ k}$.

The reward function is defined by

$$f_{w^\star}\left((w_i)_{i \in [k]}\right) = \left(\prod_{i \in [k]} g(\rho(w_{i-1}, w_i))\right) g(\rho(w_k, w^\star)),$$

$$\text{where: } g(x) = 1 - \frac{x}{p},$$

with base case $f(()) = g(\rho(w_0, w^\star))$.

While the planner plays, it chooses sequence lengths $L_t \in [K]$ and input sequences $S_t = (w_i^t)_{i \in [L_t]} \in W^{\circ L_t}$. If it chooses to stop playing, it chooses an output $S_t = (w_i^t)_{i \in [8]} \in W^{\circ 8}$ (we distinguish this

case by letting $L_t = 0$ denote that the planner has chosen to terminate). Thus the number of actions taken is $N = \min\{t \in \mathbb{N}_+ \mid L_t = 0\}$.

After any action $S_t$, the reward is given if either $\rho(w_{L_t}^t, w^\star) < p/4$ or if $L_t = K$. In both cases the reward is $\text{Ber}(f_{w^\star}(S_t))$, a Bernoulli random variable with mean $f_{w^\star}(\cdot)$. Similarly, if the planner is done (i.e. the input is $S_N \in W^{\circ 8}$), then the reward given is $R = f_{w^\star}((w_i^N)_{i \in [k^\star]})$, where $k^\star = \min\{8, \min\{k \mid \rho(w_k^N, w^\star) < p/4\}\}$.

The last thing to specify is the oracle. Here, we allow the planner to query the oracle *part-way* through a sequence. Namely, if the planner chooses to input a sequence of length $L_t < K$, then the oracle can be queried at the end of the sequence, and a second sequence of length $L_t^2 \leq K - L_t$ can be inputted. This can be repeated as many times as desired, given that the total sequence length remains $\leq K$. The oracle will simulate the expert policy from before: upon being called at a vector $w_{k,i}$, it will either return the index of the first bit which does not agree with $w^\star$, or it will return a special action indicating that all bits are correct.

We are now ready for the main theorem.

**Theorem B.7.** *Any sound planner for* CUBEGAME *which has less than* $\Omega(p/\log p)$ *oracle queries, will have a sample complexity*

$$N \geq 2^{\Omega(p \wedge K)}$$

The proof comes in 5 lemmas, two of which (Lemmas B.8 and B.10) are analogous to properties from the expert-less CUBEGAME. The other 3 lemmas are information-theoretic and are specific to the oracle setting.

First, some properties about the reward function of CUBEGAME.

**Lemma B.8** (Properties of $f_{w^\star}$). *For any* $w^\star \in W^\star, k \in \mathbb{N}, s = (w_{k'})_{k' \in [k]} \in W^{\circ k}$, *we have*

$$\frac{1}{4} \leq f_{w^\star}(()) \leq \frac{3}{4}$$

$$0 < f_{w^\star}(s) \leq \left(\frac{3}{4}\right)^{k + \mathbb{1}[\rho(w_k, w^\star) \geq p/4]}$$

*Proof.* The proof is analogous to [WSG21, Lemma 4.2], substituting our first-order $g$ function. $\square$

The following parameters will control our sample complexity:

$$n = \min\left\{\exp(p/8)p^{-x}/20 - 5, \left(\tfrac{1}{\varepsilon} - 1\right)/9.9\right\}, \quad \text{where} \tag{23}$$

$$x = \frac{p}{16}\log p, \quad \text{and} \tag{24}$$

$$\varepsilon = \left(\frac{3}{4}\right)^{K+1} \tag{25}$$

We will see that $N = \Omega(n)$ for any sound planner.

In the original game of [WSG21], the interaction protocol is captured by $(X_t, Y_t)_{t \in [N]}$, where

- $N = \min\{t \in \mathbb{N}_+ \mid L_t = 0\}$ is the interaction length
- $L_t \in [K]$ is the input length chosen,
- $S_t$ is the sequence $(w_i)_{i \in [L_t]}$ inputted, satisfying $\rho(w_{i-1}, w_i) \geq \frac{p}{4}$
- $X_t = (L_t, S_t)$,
- $U_t = \mathbb{1}\{\rho(w_{L_t - 1}, w^\star) < p/4\}$,
- $V_t = \mathbb{1}\{\rho(w_{L_t}, w^\star) < p/4\}$, and
- $Z_t = 0$ unless $V_t = 1$ or $L_t = K$, in which case $Z_t = \text{Ber}(f_{w^\star}(S_t))$,
- $Y_t = (U_t, V_t, Z_t)$.

In our case, we need the sequence $S_t$ to include every bit flip, thus $S_t = (w_{k,i})_{k \in [L_t], i \in [p]}$ We also have two new variables, namely $O_t = (o_{k,i})$ which is an indicator that the oracle was called at step $i$ of phrase $k$ and $E_t = (e_{k,i})$ which is the answer returned by the oracle. Thus our new interaction protocol is defined by $X_t = (L_t, S_t, O_t, E_t)$ and $Y_t = (U_t, V_t, Z_t)$, where $Y_t$ remains unchanged.

The planner $A$, with $n$ interactions, in the environment defined by $w^\star$, defines a distribution over the environment:

$$P_{w^\star}^{A,n}((X_t, O_t, E_t, Y_t)_t) = \prod_{i=1}^{n} p(x_i|x_{1:i-1}, o_{1:i-1}, e_{1:i-1}, y_{1:i-1})p(o_i|x_{1:i}, o_{1:i-1}, e_{1:i-1}, y_{1:i-1})p(e_i|x_i, o_i)p(y_i|x_i).$$

(26)

Note that $p(x_i|x_{1:i-1}, o_{1:i-1}, e_{1:i-1}, y_{1:i-1})$ and $p(o_i|x_{1:i}, o_{1:i-1}, e_{1:i-1}, y_{1:i-1})$ are decisions made by the planner and $p(e_i|x_i, o_i)p(y_i|x_i)$ and $p(y_i|x_i)$ are obtained by querying the environment. We define the "abstract game $(0, w^\star)$" to always yield reward 0, and which has the same oracle as environment $w^\star$. It's distribution will be written as $P_{(0,w^\star)}^A$. Let $E_n^{w^\star}$ be the event that in the first $n$ steps the planner does not hit on any vector that is close to $w^\star$:

$$E_n^{w^\star} = \cap_{t \in [n]} \left\{ t > N \text{ or } (t = N \text{ and } \min_{i \in [8]} \rho(w_i^N, w^\star) \geq \tfrac{p}{4}) \text{ or } (t < N \text{ and } \rho(w_{L_t-1}, w^\star) \geq \tfrac{p}{4} \text{ and } \rho(w_{L_t}, w^\star) \geq \tfrac{p}{4}) \right\}$$

**Lemma B.9** (A first change of measure). *For any planner $A$ and any $w^\star \in W$, we have*

$$P_{w^\star}^A(E_n^{w^\star}) \geq \tfrac{9}{10} P_{(0,w^\star)}^A(E_n^{w^\star})$$

*Proof.* It will be shown that

$$P_{w^\star}^A(E_n^{w^\star}) \geq (1-\varepsilon)^n P_{(0,w^\star)}^A(E_n^{w^\star}). \tag{27}$$

Since $1 - \varepsilon \geq 1 - \frac{1}{1+9.9n}$ by our definition of $n$ we have that $(1-\varepsilon)^n \geq (1 - \frac{1}{1+9.9n})^n \geq \lim_{n \to \infty}(1 - \frac{1}{1+9.9.n})^n > 9/10$, and thus Equation (27) implies our result.

Let $\mathcal{H}$ be the set of all possible histories $(x_t, o_t, e_t, y_t)$ of length $n$, and $E_h = E_n^{w^\star} \cap \{H = h\}$. Note that $E_n^{w^\star}$ is the disjoint union of $E_h$, so it is enough to show that for each $h$ we have:

$$\rho = \frac{P_{w^\star}^A[E_h]}{P_{(0,w^\star)}^A[E_h]} \geq (1-\varepsilon)^n,$$

for each $h$ such that $P_{(0,w^\star)}^A(E_h) > 0$. So, let $h = (x_t, o_t, e_t, y_t)$ be such that $P_{(0,w^\star)}^A(E_h) > 0$. This implies in particular that $y_t = (0, 0, 0) \, \forall t$. Now, both $P_{w^\star}^A$ and $P_{(0,w^\star)}^A$ factorize according to Eq. (26), giving:

$$\frac{P_{w^\star}^A[E_h]}{P_{(0,w^\star)}^A[E_h]} = \prod_i \frac{p(x_i|x_{1:i-1}, o_{1:i-1}, e_{1:i-1}, y_{1:i-1})p(o_i|x_{1:i}, o_{1:i-1}, e_{1:i-1}, y_{1:i-1})p^{w^\star}(e_i|x_i, o_i)p^{w^\star}(y_i|x_i)}{p^A(x_i|x_{1:i-1}, o_{1:i-1}, e_{1:i-1}, y_{1:i-1})p^A(o_i|x_{1:i}, o_{1:i-1}, e_{1:i-1}, y_{1:i-1})p^{(0,w^\star)}(e_i|x_i, o_i)p^{(0,w^\star)}(y_i|x_i)}.$$

Since we are conditioning on the same fixed history, the terms involving decisions made by the planner will cancel, and similarly the variable $E_i$ also behaves the same in both environments (since the oracle for $w^\star$ is the same). We are left with:

$$\rho = \prod_i \frac{p^{w^\star}(y_i = (0,0,0)|x_i)}{p^{(0,w^\star)}(y_i = (0,0,0)|x_i)}$$

The denominator always has probability 1 in the environment $(0, w^\star)$, and since $U_t = V_t = 0$ under the set $E_n^{w^\star}$ (the planner is never close to $w^\star$), we have $Y_t = 0 \iff Z_t = 0$, so it remains to control

$$\rho = \prod_{i=1}^{n} P_{w^\star}^A(Z_t = 0|x_t).$$

Again since the planner is never close to $w^\star$, $Z_t = 1$ only if $l_t = K$, in which case we have $P_{w^\star}^A(Z_t = 1|x_t) \geq (3/4)^{K+1} = \varepsilon$ by definition of the reward $f_{w^\star}$ obtained from reaching level $K$. $\qquad \square$

The next lemma simply bounds the number of vectors in $W$ which are close to any fixed vector.

**Lemma B.10** (Hypercube counting). *For $\tilde{w} \in W$, let $W_{close}(\tilde{w}) = \{w \in W \mid \rho(w, \tilde{w}) < p/4\}$. Then $|W_{close}(\tilde{w})| \leq 2^p \exp(-p/8)$*

*Proof.* Identical to [WSG21, Lemma 4.4]. $\qquad\qquad\square$

Recall that $E_n^{w^\star}$ is the "bad event" for the planner. We study its complement, $(E_n^{w^\star})^c$, which satisfies $(E_n^{w^\star})^c \subset \{w^\star \in Z\}$, where

$$Z := \cup_{t \in [n \wedge (N-1)]} \left(W_{\text{close}}(w_{L_t-1}^t) \cup W_{\text{close}}(w_{L_t}^t) \cup (\cup_{i \in [8]} W_{\text{close}}(w_i^N))\right),$$

i.e. the event that for some $t$ we have $\rho(w_{L_t-1}^t, w^\star) < p/4$ or $\rho(w_{L_t}^t, w^\star) < p/4$ or that for some $i \in [8]$ we have $\rho(w_i^N, w^\star) < p/4$. We define the "abstract game" $(0,0)$, where the planner has access to an oracle but, when queried, rather than giving information about the "true" $w^\star$, the oracle will simply return a uniformly random bit in $[p]$.

**Lemma B.11** (A second change of measure). *For any planner $A$ with an oracle budget of $x$, we have that:*

$$P_{0,\hat{w}}^A(\hat{w} \in Z) = p^x P_{0,0}^A(\hat{w} \in Z)$$

*Proof.* As before, consider the set of histories $\mathcal{H}$, and let $Z_h = Z \cap \{H = h\}$. Writing out the importance ratio gives:

$$\rho = \frac{P_{0,w^\star}^A[Z_h]}{P_{(0,0)}^A[Z_h]} = \prod_i \frac{p(x_i|x_{1:i-1}, o_{1:i-1}, e_{1:i-1}, y_{1:i-1})p(o_i|x_{1:i}, o_{1:i-1}, e_{1:i-1}, y_{1:i-1})p^{0,w^\star}(e_i|x_i, o_i)p^{(0,w^\star)}(y_i|x_i)}{p(x_i|x_{1:i-1}, o_{1:i-1}, e_{1:i-1}, y_{1:i-1})p(o_i|x_{1:i}, o_{1:i-1}, e_{1:i-1}, y_{1:i-1})p^{(0,0)}(e_i|x_i, o_i)p^{(0,0)}(y_i|x_i)}$$

Again, as before, all the terms involving the planner will cancel, since they are conditioned on the same history and thus the planner will make the same decisions. Similarly, in both games the reward is deterministically 0 thus the $p^{(0,w^\star)}(y_i|x_i) = p^{(0,0)}(y_i|x_i)$. We are left with

$$\rho = \prod_{i=1}^n \frac{p^{(0,w^\star)}(E_i = e_i \mid x_i, o_i)}{p^{(0,0)}(E_i = e_i \mid x_i, o_i)}$$

Note that the top probability is deterministic (since the true expert is) and is only equal to 1 at most $x$ times (recalling that $x$ is the total number of oracle calls allowed). We are simply left with the (inverse of the) probability that the random oracle returns any given answer, which is $1/p$. Thus we end up with $\rho = p^x$. $\qquad\square$

**Lemma B.12** (Finding a bad $w^\star$ for planner $A$). *For any abstract planner $A$ there exists $w^\star \in W^\star$ such that*

$$P_{w^\star}^A(E_n^{w^\star}) \geq (9/10)^2$$

*Proof.* Note that, by Lemma B.9, it is sufficient to show that $P_{0,w^\star}^A((E_n^{w^\star})^c) \leq \frac{1}{10}$. Recall that for any $\hat{w} \in W^\star$ we have $(E_n^{\hat{w}})^c \subseteq \{\hat{w} \in Z\}$, where

$$Z := \cup_{t \in [n \wedge (N-1)]} \left(W_{\text{close}}(w_{L_t-1}^t) \cup W_{\text{close}}(w_{L_t}^t) \cup (\cup_{i \in [8]} W_{\text{close}}(w_i^N))\right)$$

By a union bound and Lemma B.10 we have that $|Z| \leq (2n+8)2^p \exp(-p/8)$. Since $W^\star = W \setminus W_{\text{close}}(1) \setminus W_{\text{close}}(-1)$, this also gives that $|W^\star| \geq 2^p(1 - 2\exp(-p/8))$. We pick $w^\star$ according to:

$$w^\star = \operatorname{argmin}_{\hat{w} \in W^\star} P_{0,w^\star}^A(\hat{w} \in Z).$$

Putting things together and using Lemma B.11 gives:

$$2^p(1 - 2\exp(-p/8))P_{0,w^\star}^A(w^\star \in Z) \leq |W^\star|P_{0,\hat{w}}^A(w^\star \in Z)$$

$$\leq \sum_{\hat{w} \in W^\star} P_{0,\hat{w}}^A(\hat{w} \in Z) \leq \sum_{\hat{w} \in W} P_{0,\hat{w}}^A(\hat{w} \in Z) \leq \sum_{\hat{w} \in W} p^x P_{0,0}^A(\hat{w} \in Z)$$

$$= p^x \sum_{\hat{w} \in W} P_{0,0}^A(\hat{w} \in Z) = p^x \sum_{\hat{w} \in W} \mathbb{E}_{0,0}^A[\mathbb{1}[\hat{w} \in Z]] = p^x \mathbb{E}_{0,0}^A\left[\sum_{\hat{w} \in W} \mathbb{1}[\hat{w} \in Z]\right]$$

$$= p^x \mathbb{E}_{0,0}^A[|Z|] \leq p^x(2n+8)2^p \exp(-p/8)$$

Rearranging gives that

$$P_{0,w^\star}^A((E_n^{w^\star})^c) \leq P_{0,w^\star}^A(w^\star \in Z) \leq \frac{(2n+8)2^p \exp(-p/8)p^x}{2^p(1-2\exp(-p/8))} \leq 2(n+5)p^x \exp(-p/8) \leq \frac{1}{10},$$

where the last line followed from our bound on $n$ (Eq. (23)). $\qquad\square$

We are now ready to prove Theorem B.7. In fact, there is not much left to do.

*Proof (of Theorem B.7).* Let the planner $A$ be sound and have an expected query cost $\bar{N}$, and $w^\star$ the vector from the previous lemma. Then by Markov's inequality we have:

$$P_{(0,w^\star)}^A[N-1 \geq n] \leq \frac{\bar{N}}{n}$$

Letting $E' = E_n^{w^\star} \cap \{N-1 < n\}$ we have

$$P_{0,w^\star}^A[E'] \geq (9/10)^2 - \frac{\bar{N}}{n}$$

Under event $E'$, the output of the planner satisfies $\rho(w_i^N, w^\star) \geq p/4$ for all $i \in [8]$, so the reward at the end of the game is $R < (3/4)^9$. Combined with soundness this gives

$$\frac{1}{4} - 0.01 \leq f_{w^\star}(()) - 0.01 \leq \mathbb{E}_{w^\star}^A[R] \leq (\frac{3}{4})^9 + (1 - P_{0,w^\star}^A[E'])\frac{3}{4}$$

$$\leq (\frac{3}{4})^9 + (1 - (9/10)^2)\frac{3}{4} + \frac{\bar{N}}{n}\frac{3}{4},$$

which requires $\bar{N} > 0.02n$, namely

$$N > 0.02 \min \left\{ \exp(p/8)p^{-x}/16 - 5, \left(\frac{1}{\varepsilon} - 1\right)/7.5 \right\}$$

Lastly, note that when $x \leq \frac{p}{16 \log p}$ we have

$$\log n = \log(\exp(p/8)) - \log(p^x) = \frac{p}{8} - x\log p \geq \frac{p}{8} - \frac{p}{16\log p}\log p = \frac{p}{16},$$

thus $n = \Omega(\exp(p/16))$ and in particular $N = \Omega\{2^{p\wedge K}\}$. $\qquad\square$

To prove Theorem B.3, the last thing to show is that a learner which solves the MDP can be used to solve CUBEGAME. This reduction follows exactly as in Section 4.8 of [WSG21]

## C  On $\pi^\circ$ linearity

This section shows that, when $\pi^\circ \neq \pi^\star$, $q^\circ$ can be linear with $d-$dimensional features yet these features do not realize $\pi^\circ$-linearity.

The MDP is as follows: the states are arranged in a binary tree of length $H$. The action space is $\{\ell, \mathbf{r}\}$, corresponding to the $\ell$eft and $\mathbf{r}$ight actions. Transitions are deterministic. The reward for every $\ell$eft action is $-1$, the reward for every $\mathbf{r}$ight action is $+1$. See Figure 1.

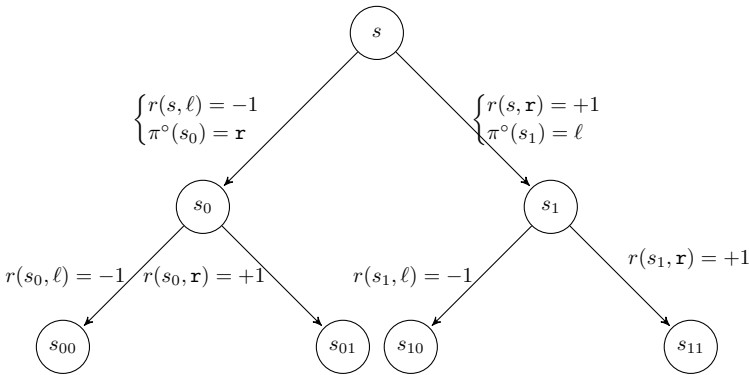

Figure 1: $q^\circ$-linearity does not imply $\pi^\circ$-linearity

Note that we can identify every state with the action sequence that led to it (with the starting state corresponding to the empty sequence). The policy $\pi^\circ$ is defined such that, if $s = (a_0, \ldots, \ell)$ then $\pi^\circ(s) = \mathbf{r}$ and otherwise if $s = (a_0, \ldots, \mathbf{r})$ then $\pi^\circ(s) = \ell$. Thus the policy will alternate the action taken at every step. This defines the $q^\circ$ function:

$$q^\circ(s, a) = \begin{cases} 0, & \text{if } h = 0 \bmod 2 \\ -1, & \text{if } h = 1 \bmod 2 \text{ and } a = \ell \\ +1, & \text{if } h = 1 \bmod 2 \text{ and } a = \mathbf{r} \end{cases}$$

Note that we can linearize this in one dimension via the features $\varphi^\circ(s, a) = q^\circ(s, a)$ and $\theta = 1$. However, these features do not linearly-realize $\pi^\circ$: since $\theta > 1$ then the argmax at every odd horizon will always be the right action (since $\varphi^\circ(s, \mathbf{r}) = 1$ and $\varphi^\circ(s, \ell) = -1$).

Note that $\pi' = \text{argmax}_a\{q^\circ(s, a)\}$, the greedy policy derived from $q^\circ$ is by definition linear with those features. For the special case where $\pi^\circ = \pi^\star$ then the greedy policy $\pi'$ lines up with the policy $\pi^\circ$, so we get linearity for free in that case.