# OpenReview forum: "A Few Expert Queries Suffices for Sample-Efficient RL with Resets and Linear Value Approximation"
_NeurIPS.cc/2022/Conference — NeurIPS 2022 Accept_

### Official Review · Reviewer_oG41 · 2022-07-01

**Rating:** 7
**Confidence:** 4
**Soundness:** 3 good
**Presentation:** 3 good
**Contribution:** 4 excellent

**Summary:**

The paper studies the RL problem of approximating the behavior of an expert demonstrator in a setting where the learner is allowed to both interactively query the environment and query the expert. Under the assumption that the value function of the expert is linearly realizable in a known feature mapping, the authors prove that the Delphi algorithm finds an eps optimal policy using O(d) samples and polynomially many samples in the other problem parameters.

**Questions:**

In L:124, the authors claim that assumption 2.1 is necessary. This isn’t supported by the theoretical results. The assumption is definitinitely sufficient, but it isn’t strictly necessary? There could be other assumptions (distinct from 2.1) that could enable sample efficient learning.

I can see that the expert policy is consistent if it corresponds to the optimal policy, but why should we expect this to hold generically for non-optimal policies?

In Theorem 3.1, the authors claim that the algorithm is computationally efficient. What is the precise, formal statement here? There is no formal runtime bound that is provided.


**Limitations:**

These are adequately adressed.

**Strengths And Weaknesses:**

The paper makes significant progress towards understanding the possibilities (and limitations) of learning a good policy in MDPs under linear function approximation. They introduce a new (and sensible) sufficient condition for the problem to be statistically tractable.

The paper is very clearly written. It nicely communicates the intuition behind the algorithm and why we might expect it to work. The lower bounds on the number of expert queries that are necessary nicely complement the analysis of the Delphi algorithm and add to the picture of learning with expert demonstrations. The robustness results regarding mispecified dynamics and value functions are also interesting.

---

> ### Author Response · Authors · 2022-08-02
> **Reply to Reviewer oG41**
>
> We thank the reviewer for their positive comments and their questions. We answer the three questions below:
>
> > "In L:124, the authors claim that assumption 2.1 is necessary [...] There could be other assumptions that could enable sample efficient learning."
>
> You are correct that the hardness could be circumvented by some other assumptions or sources of side information. What we wrote is ``Thus, any algorithm *for this setting* must necessarily make use of Assumption 2.1.'', which refers to the scenario where Assumption 2.1 is the only additional assumption one can make ("this setting"), i.e., no learner which is given Assumption 2.1, 2.2, and 2.3 can solve the problem without using Assumption 2.1. That said, your point is well taken and we admit that this might have been confusing due to lack of detailed explanation; we will revise this sentence to clarify this point.
>
> > "I can see that the expert policy is consistent if it corresponds to the optimal policy, but why should we expect this to hold generically for non-optimal policies?"
>
> We are a bit confused about this question (what does it mean for a policy to be consistent?), and are guessing that the reviewer is asking what happens when the expert policy is not optimal. (If not, please let us know). In fact, despite that we wrote the abstract in terms of an optimal expert policy (for conceptual simplicity), our actual results are robust to non-optimal experts and we did not make the assumption that expert policy is optimal (see Lines 43 and 106). Our agent can learn a policy nearly as good as any expert's policy.
>
> > In Theorem 3.1, the authors claim that the algorithm is computationally efficient. What is the precise, formal statement here? There is no formal runtime bound that is provided.
>
> As written in the outline of proof for Theorem 3.1 (cf. Line 212), our algorithm is computationally efficient since the only computationally intensive step is the optimization problem corresponding to the optimistic choice over linear parameters. As shown in Line 212, the program has a linear objective, and most of  the constraints (Line 17, Algorithm 1) are linear constraints (each absolute value constraint can be split into 2 linear constraints). The only nonlinear constraint is the $\ell_2$-norm constraint of parameters (Line 2, Algorithm 1) that characterizes the initial (unrefined) parameter space, which is convex. Taken together, this is simply a convex program, with polynomially many parameters and constraints given in standard forms, in which case the polynomial computational complexities are a well-known result and there are a plethora of efficient methods for solving such problems (e.g. by the ellipsoid method or cutting plane methods). That said, we admit that the claim of computational efficiency is a bit casual, and we will revise and add these details and a precise runtime.

---

> > ### Comment · Reviewer_oG41 · 2022-08-07
> > **Thank you for the detailed response!**
> >
> > My question was regarding why we should expect the expert policy should be Bellman consistent. I think I may have misread the exposition the first time around, this makes sense now. Thank you for the detailed responses.

---

### Official Review · Reviewer_v8cV · 2022-07-08

**Rating:** 6
**Confidence:** 3
**Soundness:** 3 good
**Presentation:** 3 good
**Contribution:** 2 fair

**Summary:**

Recent work has shown that linear realizability of the optimal value function is not a sufficient condition for sample-efficient RL (i.e. you need exponentially many samples to find a near optimal policy). This work shows that it is possible to learn a near-optimal policy using only polynomially many samples if the learner is also given access to an oracle it can query and which will return a “good” action at a given state. In particular, they show that if the value function of this oracle can be parameterized by a $d$-dimensional feature vector, then with $O(d)$ calls to the oracle, it is possible to learn a near-optimal policy using only polynomially many samples. In addition, a lower bound is given which shows that at least $\Omega(\sqrt{d})$ oracle queries are necessary for sample-efficient learning.

**Questions:**

None.

**Limitations:**

Limitations are discussed. No discussion is given on societal impact. While the work is primarily theoretical and may not have obvious immediate societal impact, it is contributing to the larger field of RL which does have practical impact. Given this, I would encourage the authors to consider what such impacts might be, and how negative impacts might be mitigated.

**Strengths And Weaknesses:**

Strengths:
- Understanding when sample-efficient learning in RL is possible is an important question that has been of much interest in recent years. This work makes a non-trivial contribution to developing our understanding of this. I believe it is interesting and not obvious that by making limited oracle queries a previously intractable problem becomes tractable, though I have some concerns on the rate (see below).

Weaknesses:
- I believe the primary weakness of this work is the gap between the $O(d)$ upper bound on the oracle queries, and the $\Omega(\sqrt{d})$ lower bound. It is easy to construct settings where $O(d)$ calls to an oracle trivially solves the problem. For example, if we consider a tabular MDP with $S$ states and $A$ actions, the typical mapping from tabular to a linearly parameterized MDP has dimension $d = SA$. In this setting, if we query an oracle $SA$ times, that will simply give us the oracle policy, and the problem is solved. A similar conclusion can be reached in linear MDPs. Even in the more difficult setting considered in this work, the proposed algorithm seems to be doing something similar: since it assumes the oracle policy has a linearly realizable value function, it essentially just queries it in a spanning set of directions, which allows the value function to be approximately recovered. It does not seem too surprising this is possible, but doing this in $O(\sqrt{d})$ oracle calls seems much less trivial, as in that case the learner would have to do a significant amount of learning without relying on the oracle. Thus, I think closing this gap between $O(d)$ and $\Omega(\sqrt{d})$ is essential for this to be a convincing result, but I don’t think the style of algorithm proposed here is able to do this.
- One suggestion for improving this would be to restrict the learner so it can only access the oracle in the state it is currently visiting. The proposed algorithm only requires this type of access, and it would make it at least somewhat less trivial to solve easier problems with an oracle, as some exploration would still be required.
- Is the assumption that the oracle policy is linear in the features necessary? It seems essential to the algorithm. I don’t think this is a critical issue since, if we take the oracle policy to simply be the optimal policy, this will be satisfied by our linear realizability assumption, but relaxing this assumption would still strengthen the results.

---

> ### Author Response · Authors · 2022-08-02
> **Reply to Reviewer v8cV**
>
> We thank the reviewer for reading the paper and for their thoughtful feedback. We address their comments below.
>
> > "It is easy to construct settings where $O(d)$ calls to an oracle trivially solves the problem... but doing this in $O(\sqrt{d})$ oracle calls seems much less trivial... I don’t think the style of algorithm proposed here is able to do this."
>
> **Response**: We agree that closing the gap between the upper bound and lower bound is an interesting question but respectfully disagree that ``$O(d)$ calls to an oracle trivially solves the problem''. The reviewer's intuition seems to be that one can find/compute a spanning set of features and then just query the oracle policy on states with those features. The first problem is that, by this, the reviewer is implicitly assuming a powerful generative model where the learner has access to all the states and their features, and can visit the states corresponding to the computed spanning set. This is substantially stronger than our setup that only allows resetting to the previous state. Even with a generative model, *computing* a spanning set of features requires pre-screening the entire set of features and incurs $O(|\mathcal{S}|)$ computation. Without a generative model (as in our case), learning *how to get to* those states requires at least $O(|\mathcal{S}||\mathcal{A}|)$ samples even in the simplest case of deterministic dynamics. Therefore, while the reviewer's intuition does capture part of what the algorithm is doing, this intuition ignores a very important part of the challenge, namely how to find the spanning features without prescreening the state space, and instead via interacting with the MDP from its start state. This requires clever exploration strategies and addressing exploration and learning (i.e., forming TD-error based constraints and maintaining version spaces) in an intertwined manner.
>
> The second issue with this claim is that it is not even clear *what to do* with this spanning set of features (and in fact this ends up being the wrong object to look at, as explained shortly). Estimating the value of $v^\circ$ at each of the basis vectors would require collecting rollouts from those states, which will introduce a factor of $H$ in the number of oracle queries. On the other hand, the policy $\pi^\circ$ itself does not need to be linear (despite that $v^\circ$ is, see Appendix C for an example), so a classification-based approach (e.g. Behaviour Cloning) using the set of linear classifiers would fail. Our algorithm instead finds a set of state-action pairs where the *Temporal difference (TD) errors* (which we represent as vectors) span orthogonal directions (which is different from saying that the features themselves are orthogonal). This "local fitting" approach is novel to the IL literature and avoids the factors of $H$ and $1/\varepsilon$ from previous works. Even with access to the feature map and a generative model, this set of TD vectors cannot be computed in advance (since estimating the TD error requires interacting with the environment).
>
> The reviewer seems to further assume that that an upper bound of $O(\sqrt{d})$ is possible (i.e. the lower bound is tight), and our "intuitive" strategy cannot achieve this upper bound (which is seen as a weakness). It is very plausible that the looseness is on the lower bound side, since the literature's current understanding of the hardness of RL with linear value functions is not very mature. Despite several results in this space ([WAS21,WWK21,WSG21]), these all use essentially the same "mechanism" to create the exponential lower bounds. We are fairly certain this mechanism cannot give better than $\Omega(\sqrt{d})$ (as explained in Section 4), but this is not to say that the lower bound is unimprovable. Thus, a novel construction (over the already complicated one in the lower bound theorem) would be needed to improve the lower bound.
>
> > One suggestion for improving this would be to restrict the learner so it can only access the oracle in the state it is currently visiting.
>
> **Response**: The assumption that the oracle can only be queried at current states was implicit in our work. We will clarify this restriction as it does indeed strengthen our results, and thank the reviewer for this suggestion.
>
> > Is the assumption that the oracle policy is linear in the features necessary? [...] relaxing this assumption would still strengthen the results.
>
> **Response**: Perhaps there is some confusion. For the upper bound, we assume that the oracle's value function is linear in the features, but do not further need to assume that the policy itself is linear. In fact, it is possible that the policy itself takes a very complicated form (which we cannot represent), but its value function is still linear in the given basis (again see Appendix C). The additional lower bound of Theorem 4.3 does study the linear policy case, but this assumption is not the same setting as the upper bound.

---

> > ### Comment · Reviewer_v8cV · 2022-08-08
> > **Updated review**
> >
> > After reading the author's reply, I believe some of my concerns have been addressed and I will increase my score accordingly. I would encourage the authors to make more explicit what the exact query model they are using is (e.g. that they can only query the oracle in the state they are currently in) as this would help avoid some trivial cases where the learner can immediately solve an MDP with $d$ oracle calls.
> >
> > One follow-up question: I may not have been completely clear in my last point on the assumption that the oracle policy is linear—what I had meant by this is whether or not the assumption that the oracle policy's _value function_ is linear is necessary (i.e. Assumption 2.2). One could imagine a case where the optimal policy in the MDP does have a linear value function, but the oracle policy we have access to does not. From what I understand, the proposed algorithm would not work in this case?

---

> > > ### Author Response · Authors · 2022-08-09
> > > **Thanks for your reply.**
> > >
> > > We thank the reviewer for their reply.
> > >
> > > >  I would encourage the authors to make more explicit what the exact query model they are using is
> > >
> > > Will do!
> > >
> > > > One could imagine a case where the optimal policy in the MDP does have a linear value function, but the oracle policy we have access to does not. From what I understand, the proposed algorithm would not work in this case?
> > >
> > > You are correct:  our algorithm and analysis do not handle this case.
> > >
> > > That said, we want to re-iterate that we can handle the following special case:
> > > * 1) Oracle policy is optimal, and
> > > * 2) Optimal policy has a linear value function.
> > >
> > > This is a natural setting (which is what we mention in the abstract), and our setup (arbitrary oracle policy + oracle policy has linear value function) is a strict relaxation of it. When the oracle policy is no longer optimal, intuitively we can only hope to compete with the oracle policy. Competing with the actual optimal policy would require solving exploration almost from scratch since the oracle policy may be arbitrarily poor, which is intractable. In this case, the optimal policy of the MDP is no longer relevant to our learning goal, so it makes sense that the realizability assumption (#2) needs to be linked to the oracle policy, not the optimal policy.

---

### Official Review · Reviewer_TmTo · 2022-07-10

**Rating:** 5
**Confidence:** 3
**Soundness:** 1 poor
**Presentation:** 2 fair
**Contribution:** 2 fair

**Summary:**

Assuming that only the optimal value function is linearly-realizable for the homogeneous finite horizon MDP, this paper proposes a sample-efficient algorithm with access to interactive demonstrations from an expert policy.

**Questions:**

The authors exaggerate their results due to misunderstanding others' conclusions.

**Limitations:**

The authors exaggerate their results due to misunderstanding others' conclusions.

**Strengths And Weaknesses:**

The authors exaggerate their results due to misunderstanding others' conclusions.
Specifically, the exponential lower bounds (w.r.t. $H$, $d$) in [WAS21] and [WWK21] are constructed based on exponentially large action spaces.
As far as I know, the prior arts only establish the $\mathrm{poly}(d, H, A, 1/\varepsilon)$ lower bound.
The authors should correct their claim throughout the paper, if they cannot provide the exponential lower bounds (w.r.t. $H$, $d$, $A$).

In addition, it seems a better result is given in [AJKS19] for the infinite horizon MDP, which I see in Section 5.
If this is right, the authors should discuss the difficulties of this extension.

The writing should be improved due to many typos.

---

> ### Author Response · Authors · 2022-08-02
> **Reply to Reviewer TmTo**
>
> We thank the reviewer for reading the paper and providing feedback. We respond to their comments below.
>
> > The authors exaggerate their results due to misunderstanding others' conclusions...the exponential lower bounds (w.r.t. $H$, $d$) in [WAS21] and [WWK21] are constructed based on exponentially large action spaces. [...] The authors should correct their claim throughout the paper, if they cannot provide the exponential lower bounds (w.r.t. $H$, $d$, $A$).
>
> **Response**: The reviewer has made a technical mistake. While it is true that the lower bound constructions in [WAS21] and [WWK21] have exponentially large action spaces, the one in [WSG21] (discussed at several points in our paper e.g. Lines 33, 264, 305, 350) **only has polynomially many** actions (in fact fewer than $d$) in its construction. Therefore, $poly(d, H, A)$ is indeed ruled out by existing results. Our lower bound, which builds on top of the construction of [WSG21], therefore *also* precludes any algorithm with $poly(d,H,A)$ exploration budget to solve this problem without $\Omega(\sqrt{d})$ expert queries.
>
> > In addition, it seems a better result is given in [AJKS19] for the infinite horizon MDP, which I see in Section 5. If this is right, the authors should discuss the difficulties of this extension.
>
> **Response**: By Section 5 does the reviewer mean Section 5 of our paper or Section (Chapter) 5 of [AJKS19]? Chapter 5 of [AJKS19] is not related to problem setting described here so we will assume that the reviewer meant our Section 5, which is the Related Works section. We are not sure which result from our Section 5 the reviewer thinks is better than ours. All prior works, and in particular those we have discussed, have worse oracle complexities than our method. The result from [AJKS19] which we quote is that Behaviour Cloning and AggraVaTe have oracle complexities of $O(dH^4/\varepsilon^2)$ (after translating from the discounted setting), while ours is $O(d \log(1/\varepsilon))$. We also contacted the authors of [AJKS19] regarding this point and they were equally unsure about the reviewers' comment.
>
> > The writing should be improved due to many typos.
>
> **Response**: Thank you for pointing out the typos. We will review the text and fix them.

---

### Official Review · Reviewer_T4Ki · 2022-07-11

**Rating:** 7
**Confidence:** 3
**Soundness:** 4 excellent
**Presentation:** 4 excellent
**Contribution:** 3 good

**Summary:**

This paper studies the problem of reinforcement learning of reinforcement learning with a linearly realizable optimal value function. While this problem is infeasible in general (as shown by recent lower bounds), this paper shows that two additional assumptions are sufficient for sample efficiency in this setting: O(d) queries to an expert and the ability to resample arbitrarily many times from the most recent state visited. A new algorithm is proposed that achieves poly sample complexity under these assumptions. Lower bounds are shown justifying at least O(\sqrt{d}) queries from the expert.


**Questions:**

- While the paper’s theoretical contributions are already very strong, I am curious why there were no experiments, especially considering the algorithm is advertised as being computationally efficient? How do you think this algorithm will compare to IL algorithms like BC, Dagger, and Aggrevate? How would it compare to RL algorithms?
- I’m also curious about how necessary Assumption 2.3 (resets) is. It is clear that it plays an important role in the proposed algorithm to identify states for which there is consistency. What seems to be the technical challenges needed to remove this?
- Can you also comment on to what extent using Assumption 2.3 might be helping with the linear realizability assumption? For example, in some offline RL papers, one sometimes sees that double-sampling enables sample-efficient guarantees even with few assumptions on the function class (like completeness), so I am curious if something similar is happening under the hood here. Clearly resetting is not enough by itself given the existing lower bounds, but I am curious to know how much it helps / if it's necessary.



**Limitations:**

The limitations are adequately addressed.

**Strengths And Weaknesses:**

- In my view, the results constitute contributions to both reinforcement learning (specifically to the study of weaker forms of realizability) and imitation learning. On the RL side, we learn that a notoriously harder problem is solvable with a limited amount of side information in the form of expert actions. On the IL side, we get a clearer picture of how much interactive expert information is actually necessary to learn in a sample efficient manner.
- The paper is very well written and easy to understand. It does a good job of placing itself in the literature and making clear what is and is not possible with existing results.
- The insight of how to use the expert queries is interesting and seems to be new.

Weaknesses
- The algorithm requires a resetting assumption (this is weaker than a full generative model but stronger than the standard episodic RL setup)
- There is a gap between the sqrt{d} lower bound and d upper bound.

---

> ### Author Response · Authors · 2022-08-02
> **Reply to Reviewer T4Ki**
>
> We thank the reviewer for their positive comments and their questions. We address the three questions below.
>
> > While the paper’s theoretical contributions are already very strong, I am curious why there were no experiments, especially considering the algorithm is advertised as being computationally efficient?
>
> **Response**: As the reviewer recognized, this is a theoretical paper, and we believe the theoretical contributions alone are sufficient for publication. Empirical experiments  are indeed an interesting future direction, but few RL algorithms with rigorous guarantees can be implemented as-is in complex environments, and often need nontrivial adaptations to implement, which we consider out of the scope of our current work. The computational efficiency of our algorithm is a rigorous mathematical claim about its computational complexity (cf. our response to Reviewer oG41 about precise statement of computational efficiency). There are many theoretical RL works that propose computationally efficient algorithms (esp. in online RL for tabular and linear MDPs), but most of those papers do not come with experimental results.
>
> > Can you also comment on to what extent using Assumption 2.3 might be helping with the linear realizability assumption? For example, in some offline RL papers, one sometimes sees that double-sampling enables sample-efficient guarantees even with few assumptions on the function class (like completeness), so I am curious if something similar is happening under the hood here.
>
> **Reponse**: First of all, the reviewer seems to suggest that assuming completeness is weaker than realizability (``few assumptions (like completeness)''). The relationship is the other way around: completeness is much stronger and implies realizability; see page 4 of [CJ19]. Therefore, we are operating under a substantially weaker assumption. As the reviewer correctly points out, a major reason why offline RL often needs completeness is because of the double-sampling difficulty that precludes direct estimation of the Bellman error of candidate functions. The role of resets is similar -- we use these (in part) to estimate the TD errors of candidate actions (by sampling the same state-action pair multiple times), and thus avoid relying on completeness and only need realizability.
>
> The additional structure of linearity (which is not needed in offline RL if one has completeness [CJ19]) interplays with the reset function in that we use the reset function to check consistency, and the linear property of the optimal value function to guarantee that a certain number of consistency checks on well-chosen states will eventually *extrapolate* to "global consistency". Without linearity, one might have to check that a candidate value function is consistent at *every* state before being able to guarantee that it is the optimal value function (e.g. picture a large binary tree with a single rewarding leaf node; a hypothesis of $\hat v=0$ satisfies consistency at all but one state).
>
> > I’m also curious about how necessary Assumption 2.3 (resets) is. [...] What seems to be the technical challenges needed to remove this?
>
> **Response**: As you point out, the reset assumption is needed to identify which actions are consistent at a given state, and this is achieved by trying out all actions in this state. (This is a second use of resets which is complementary to what we discussed above, i.e., trying all actions at a given state vs. trying the same state-action pair multiple times.) The technical challenge would thus be to find a way to identify the most consistent action *without explicitly trying* all of them. This seems like a highly non-trivial extension -- one speculative idea would be to have an additional classifier or adversarial bandit learning said action (similar to the $V$-learning paper of Jin, Liu, Wang, and Yu [2022]), though it is not a priori clear how to do this in a non-tabular way (i.e. without having one classifier per state).

---

> > ### Comment · Reviewer_T4Ki · 2022-08-08
> > **Reply**
> >
> > Thanks for the response. I've read it and the other reviews. No more questions at this time.

---

### Meta-Review · Area_Chair_Wuhq · 2022-08-21

**Recommendation:** Accept
**Confidence:** Certain

**Metareview:**

We thank the authors for their submission.

The paper studies finite-horizon MDPs in which *only* the optimal value function is realized by a linear function. The learner has access to the MDP via a generative model and has to minimize its sample complexity for finding a policy with approximately optimal value function. This was shown by prior work to require a number of samples exponential in the problem parameters.
Contributions: First, an algorithm that guarantees polynomial sample complexity if the learner has additional access to $O(d)$ expert demonstrations. Second, a lower bound of $\Omega(\sqrt{d})$ on the required number of expert queries.
The work adds to our understanding of when MDPs with linear function approximation are solvable, showing that a hard RL problem becomes easy with a small amount of additional information. It is very well-written.

**Award:**

No

---

### Decision · Program_Chairs · 2022-09-14

Accept